statistics

online estimates, disease atlas, small area estimates, cancer atlas, cancer incidence, geographical patterns

**Author for correspondence:**
Farzana Jahan
e-mail: f.jahan@hdr.qut.edu.au

# Augmenting disease maps: a Bayesian meta-analysis approach

Farzana Jahan[1], Earl W. Duncan[1], Susanna M. Cramb[2], Peter D. Baade[2] and Kerrie L. Mengersen[1]

[1]School of Mathematical Science, ARC Centre of Excellence for Mathematical and Statistical Frontiers, Queensland University of Technology, Queensland, Australia
[2]Cancer Council Queensland, Brisbane, Queensland, Australia

FJ, 0000-0002-0300-6315; EWD, 0000-0002-5146-7810;
SMC, 0000-0001-9041-9531; PDB, 0000-0001-8576-8868;
KLM, 0000-0001-8625-9168

Analysis of spatial patterns of disease is a significant field of research. However, access to unit-level disease data can be difficult for privacy and other reasons. As a consequence, estimates of interest are often published at the small area level as disease maps. This motivates the development of methods for analysis of these ecological estimates directly. Such analyses can widen the scope of research by drawing more insights from published disease maps or atlases. The present study proposes a hierarchical Bayesian meta-analysis model that analyses the point and interval estimates from an online atlas. The proposed model is illustrated by modelling the published cancer incidence estimates available as part of the online Australian Cancer Atlas (ACA). The proposed model aims to reveal patterns of cancer incidence for the 20 cancers included in ACA in major cities, regional and remote areas. The model results are validated using the observed areal data created from unit-level data on cancer incidence in each of 2148 small areas. It is found that the meta-analysis models can generate similar patterns of cancer incidence based on urban/rural status of small areas compared with those already known or revealed by the analysis of observed data. The proposed approach can be generalized to other online disease maps and atlases.

## 1. Introduction

A major field of research in spatial epidemiology involves the construction of disease maps showing the spatial distribution of disease [1]. Disease mapping is usually undertaken using observational disease data (spatially aggregated count data) to show small area estimates of disease incidence, prevalence, mortality or relative risk of a specific disease in small areas [2].

Sometimes the disease maps are published in the form of an atlas, for example, Surveillance Atlas of Infectious Diseases [3], the Atlas of Heart Disease and Stroke [4], the Environment and Health Atlas of England and Wales [5], the U.S. Atlas of Cancer Mortality [6], Atlas of Cancer in Queensland [7], Cancer Atlas of the United Kingdom and Ireland [8] and Australian Cancer Atlas [9]. Since the observational disease data are usually subject to privacy and confidentiality constraints, the modelled disease rates for small areas of a country or region are reported in the atlases. If these reported estimates can be used to extract important information about the spatial distribution of a disease and the influence of different risk factors on the incidence and/or survival, this would widen the scope of research in spatial epidemiology.

Meta-analysis is a popular method for combining results from different studies quantitatively [10], particularly when those results are in the form of aggregate or summary data [11,12]. The present study aims to develop a Bayesian hierarchical meta-analysis model which will model published estimates of disease from atlases to identify specific patterns in disease incidence and will yield similar types of inferences to those obtained using the raw data. The proposed model is illustrated for a specific disease, cancer, in the present study.

Cancer is the second leading cause of death in the world [13] and is an important topic of research in the field of spatial epidemiology [14]. In the twenty-first century, cancer is expected to become the single most significant obstacle to increasing life expectancy in every country in the world [15]. In 2018, the estimated number of new cases of cancer was 18.1 million and estimated number of deaths from cancer was 9.6 million worldwide [15].

A major field of research in cancer epidemiology is the assessment of spatial patterns of cancer. Substantial research has focused on assessing spatial patterns of cancer in different parts of the world [16–23]. The common sources of spatially referenced cancer data used for spatial analysis of cancer are population-based cancer registries; those are supplemented with additional population data, health survey data along with environmental and remote sensing data [24]. Most of the previous studies modelling geographical patterns of cancer incidence used observed population-based counts of new cases of cancer in a region or area, adjusted by the size of the population in each area, for specified periods obtained from a population or cancer registry [20,25,26].

Models for spatial patterns often incorporate spatial smoothing. Among many methods of smoothing, Bayesian methods [27] are very popular, for example, via Gaussian Markov Random fields [28], especially conditional autoregressive (CAR) models, as a prior for the spatial effects [29]. Some common model formulations using CAR representations for spatial smoothing in disease mapping are the Besag–York–Mollié (BYM) model [30], Leroux model [31] and Cressie model [32].

Often, researchers wish to undertake further analysis of disease patterns in these atlases. As an alternative to accessing the underlying observational data, the reported estimates in the atlases can be further modelled using a Bayesian hierarchical meta-analysis model to gain additional insights. In the literature, there are many studies that perform meta-analysis of disease estimates (standardized incidence or mortality ratio, SIR/SMR) across different published studies [33–36]. However, meta-analysis models using estimates derived from areal data from a single database, particularly to analyse the outcomes of a cancer atlas have not been explored yet, to the best of our knowledge.

The proposed Bayesian hierarchical meta-analysis model approach is not a traditional application of meta-analysis to estimates from different published studies; rather, it is an application of meta-analysis to estimates from a single study, that quantifies the ecological effect of covariates using model-based estimates. The proposed model is applied to analyse the spatially smoothed point and interval estimates available in the Australian Cancer Atlas (ACA) for the 20 different cancers in 2148 small areas (Statistical Area level 2, SA2, defined by Australian Bureau of Statistics (ABS), Australian Statistical Geography Standard (ASGS) 2011 boundaries [37]) across the country. The primary question of interest that is used to illustrate the approach is focused on whether cancer incidence varied in urban, regional and remote areas across Australia. Each of the SA2s belongs to each of the three remoteness categories according to geographical and accessibility to services within each area.

In cancer epidemiology, studies to examine the relationship of cancer incidence/survival/mortality and geographic remoteness is well researched [38–41]. For example, the relationship between risk of advanced colorectal cancer incidence in Queensland and geographical remoteness was found to be significant for those diagnosed with colon cancer [42]. A classification tree approach has also confirmed a significant association between remoteness and the incidence of several cancers [41]. The cancer disparities in different remoteness categories have also been researched in [43–46], etc. All the mentioned studies focused on the influence of remoteness on cancer outcomes using population-based cancer data. Hence, in the present study, this well-researched and important research question is chosen to provide additional comparisons by which the validity of the proposed approach can be assessed.

The meta-analysis approach was proposed because this opens up other avenues for extracting insights from data when only the summary data are published and the original data are unavailable. This is often the case for health data which are subject to privacy and confidentiality. It is true that the original authors could be approached to undertake the follow-on analyses, but this may not always be possible: for example, they could simply refuse, or not have time or funding to implement the request. Moreover, even if the original data were available, the analysis of primary data often requires some domain knowledge. The proposed approach provides a statistically valid methodology to model the published point estimates, taking into account their associated uncertainty, in straightforward manner. We show that this can facilitate new insights, in our case an enhanced understanding of the spatial distribution of cancer.

Following this introduction the paper is organized as follows: §2 consists of a description of data; §3 describes the methods and §4 reports the results; §5 outlines some possible model extensions followed by a discussion in §6.

## 2. Data

The present study uses publicly available small area estimates from the Australian Cancer Atlas (ACA) [9]. The ACA is a freely accessible and interactive online platform, showing the spatial variation in standardized incidence and excess deaths for 20 cancers across Australia (see table 13 for a list of the cancers). A key feature of the ACA is the use of Bayesian spatial models to generate the point estimates and their 95% credible intervals for the estimates of standardized incidence ratios (SIR) and excess hazard ratios (EHR) for each cancer in each of 2148 geographical areas (SA2) covering Australia. To generate the estimates of SIR, unit-level data on each cancer over different time periods were modelled using Bayesian spatial models. For 14 cancer types (oesophageal, stomach, liver, pancreatic, cervical, uterine, ovarian, kidney, brain, thyroid, non-Hodgkin lymphoma, leukaemia, myeloma and head and neck), data on a 10-year time period (2005–2014) were used. For the remaining six cancer types (bowel, lung, melanoma, breast, prostrate, all cancers combined), data on a 5-year time period (2010–2014) were used.

Statistical Areas level 2 (SA2) are medium-sized general-purpose areas designed to represent a community that interacts together socially and economically [37]. There are 2196 SA2 regions (ASGS 2011 Boundaries [37]) covering the whole of Australia without gaps or overlaps. In the ACA, SA2s with zero/nominal population, far-flung islands are excluded.

To address the research question, we focused on the SIR and information on the remoteness status in 2011 of each SA2. This information on remoteness was obtained using the Remoteness Index provided by the Australian Bureau of Statistics (ABS), which classifies small areas in Australia into five categories of remoteness based on their relative access to services. There are five categories of Remoteness—Major city, Inner regional, Outer regional, Remote and Very remote. The original five categories were combined into three classes as: 1 = Major Cities (1242 SA2s), 2 = Inner/Outer Regional (810 SA2s) and 3 = Remote/Very Remote (96 SA2s) in the present study.

## 3. Methods

### 3.1. Bayesian meta-analysis Model

Meta-analysis can be accomplished by applying fixed effects or random effects models to analyse aggregate or summary data or individual data published in different studies on the same subject [11,12]. There has been an active literature on Bayesian approaches of meta-analysis of different types of data, since the use of hierarchical Bayesian model to cast a random effects model [47]. For more details on meta-analysis using Bayesian inference, choice of prior distributions, Bayesian computation and interpretation with examples, see Koricheva *et al.* [48].

The Bayesian spatial model used to generate the estimates reported in the ACA is described in §3.2. Since this model has already incorporated spatial smoothing to protect the identity of the individuals, additional smoothing terms are not considered in the proposed meta-analysis model, although we revisit this in the section on Model extensions. In the following, we adopt the approach taken in the ACA and model each cancer individually.

#### 3.1.1. Model specification

The data available for modelling are the published estimates of the posterior mean standardized incidence rate (SIR) and corresponding 95% credible intervals for each of $N = 2148$ statistical areas (SA2). Using notation based on [49], let $Y_{i[j]}$ be the modelled log(SIR) for the $i$th SA2 ($i = 1, 2, ....2148$)

in the $j$th remoteness region ($j = 1, 2, 3$, $1 =$ major cities, $2 =$ regional and $3 =$ remote). Similarly, let $S_{i[j]}$ be the associated standard deviation of $Y_{i[j]}$, obtained from $(\log(\text{UCL}) - \log(\text{LCL}))/(2 \times 1.96)$, where LCL and UCL are, respectively, the lower and upper bounds of the corresponding published 95% credible intervals for the SIR. The proposed hierarchical Bayesian meta-analysis model for each of the cancers can be formulated as

$$Y_{i[j]} \sim N(\mu_{i[j]}, \sigma_{i[j]}^2), \quad \text{for } i = 1, 2, 3, \ldots, 2148 \text{ and } j = 1, 2, 3, \tag{3.1}$$

where $\mu_{i[j]}$ denotes the true value of the log(SIR) in the $i$th SA2 inside the $j$th region with associated variance $\sigma_{i[j]}^2$. The standard deviations $S_{i[j]}$ are used to formulate prior for the associated variance parameter, $\sigma_{i[j]}^2$, which is shown in equation (3.5). Now, $\mu_{i[j]}$ can be modelled as

$$\mu_{i[j]} \sim N(\theta_j, \sigma_j^2), \quad \text{for } i = 1, 2, 3, \ldots, 2148 \text{ and } j = 1, 2, 3, \tag{3.2}$$

where $\theta_j$ is the region-specific mean of the logSIR for the specific cancer and $\sigma_j^2$ is the corresponding variance. The region-specific mean $\theta_j$ can in turn be modelled as

$$\theta_j \sim N(\mu_0, \sigma_0^2), \quad \text{for } j = 1, 2, 3, \tag{3.3}$$

where $\mu_0$ is the overall mean log(SIR) and $\sigma_0^2$ is the associated variance.

Priors of the above hierarchical Bayesian meta-analysis model can be specified as:

$$\mu_0 \sim N(0, 100), \tag{3.4}$$

$$\sigma_{i[j]}^2 \sim \frac{\nu S_{i[j]}^2}{\chi_\nu^2}, \tag{3.5}$$

$$\tau_j \sim N^+(0, 10) \tag{3.6}$$

and

$$\tau_0 \sim N^+(0, 100), \tag{3.7}$$

where $\nu$ is the associated degrees of freedom for $\sigma_{ij}^2$, $\tau_j = 1/\sigma_j^2$ and $\tau_0 = 1/\sigma_0^2$ are the precision parameters associated with the region-specific variance and the overall variance, and $N^+$ denotes the zero-mean truncated normal distribution truncated from below at zero.

The above model is fitted for each sex and each of the 20 cancers independently. Thus, we did not include any cancer-specific or sex-specific indices.

The choices of priors and associated hyperparameters were made on the basis of the results of the sensitivity analysis. A total of 76 combinations of priors using different distributions and different values for the hyperparameters were compared and the above model was evaluated to perform reasonably well in terms of convergence (assessed by visual diagnostics and the Gelman–Rubin statistic [50]) and posterior predictive checks (posterior predictive distribution plots with observed value for summary measures, Bayesian posterior predictive $p$-values and visual comparisons between replicated data drawn using a posterior predictive distribution with the estimates from the ACA). For the prior on the overall mean, $\mu_0$, a normal prior with many different choices of variance parameter were compared. For the variance parameters, $\sigma_j^2$ and $\sigma_0^2$, after comparing all the conventional priors (inverse Gamma, half normal for standard deviation and so on), we found that, for modelling the atlas output data, placing half normal priors on precision parameters gave better model performance, hence we made the above choices. We set $\nu = 2$, as a small value of $\nu$ reflects little *a priori* knowledge about the within-study variation. For more details of the sensitivity analysis, see the electronic supplementary material.

### 3.1.2. Model implementation

The proposed model was fitted using the R programming language (R 3.5.3) [51] using the package R2jags v. 0.5-7 [52]. The MCMC output of the JAGS model was summarized in R using the *coda* package [53]. The JAGS code for the model is given in appendix A.

Three parallel MCMC chains, each with 500 000 iterations with a burn in period of 400 000 iterations were run to fit the proposed model. Convergence was examined by using visual diagnostics for the parameters of interest. The parameters for which posterior samples are drawn are: $\mu_{ij}$, $\sigma_{ij}^2$, $\theta_j$, $\tau_j$, $\mu_0$ and $\tau_0$. In this study, we focused on the region-specific estimates of mean log(SIR) represented by $\theta_j$, the corresponding precision parameters $\tau_j$ and the overall mean $\mu_0$ along with the corresponding precision parameter $\tau_0$. The region-specific means of log(SIR), $\theta_j$, or $e^{\theta_j}$, which is the mean SIR in $j$th region, are the key parameters of interest to describe the pattern of incidence of each cancer in major cities, regional and remote areas.

### 3.1.3. Model inferences

A range of measures were computed in order to identify patterns of cancer incidence by remoteness of the regions. Using the posterior estimates of region-specific means of log(SIR), $\theta_j$, in each MCMC iteration the regions were ranked in descending order and the summary of ranks along with 95% credible intervals and probability distributions of ranks for each region were calculated for each cancer. The posterior means of $\theta_j$ ($j = 1, 2, 3$) were also used to obtain pairwise comparisons of region-specific incidence of each cancer probabilistically. For each of the cancers, the values of $\theta_j$ in each of the MCMC iterations were ordered, thus enabling calculation of the probabilities of major cities having higher posterior means of log(SIR) than that of regional and remote areas, also the probability of regional areas having higher incidence than those in remote areas for males and females. These probabilities provided additional support to the comparisons made using ranks for the observed patterns of cancer incidence by remoteness. In addition to the pairwise probabilistic comparison of major cities, regional and remote areas, the point and interval estimates of posterior mean differences were calculated and reported for each of the three pairs (major cities and regional, major cities and remote, regional and remote) for all the cancers by sex.

## 3.2. Model validation

The results of the proposed Bayesian meta-analysis model for all 20 cancers were verified by comparing with those obtained from unit-level analysis of cancer registry data from each cancer in 2148 SA2s in Australia. Following the choices adopted in ACA, the Leroux model [31] was fitted using the observed incidence data for each cancer. Let $Z_{ij}$ be the number of reported cancer diagnoses in the $i$th SA2, and $j$th region modelled as

$$Z_{ij} \sim \text{Poisson}\,(E_{ij}e^{\mu_{ij}}), \quad i = 1, 2, \ldots, 2148\ j = 1, 2, 3, \tag{3.8}$$

where $E_{ij}$ represents the expected number of cancers in the $i$th SA2 and $j$th region, which can be calculated for each SA2 in each region by multiplying the SA2 population with the ratio of number of cancers diagnosed in Australia to the Australian population in each 5-year age group, then summing over all age groups. Under this model, the log(SIR), $\mu_{ij}$, was modelled as,

$$\mu_{ij} = \log\,(E_{ij}) + \beta_j I_j + R_i, \tag{3.9}$$

where $I_j \in (0, 1)$ is an indicator for the $j$th region ($j = 1, 2, 3$ for major cities, regional and remote areas, respectively) and $\beta_j$ is the associated coefficient. Note that the model adopted in the ACA does not have the covariate $I_j$ to consider the remoteness of an area in the model; it was included here in order to validate the proposed meta-analysis model.

The term $R_i$ accounts for spatial autocorrelation between the SA2s. The spatial random effects term has a conditional distribution [31];

$$R_i | R_{-i} \sim N\left(\frac{\rho \sum_k w_{ik} R_k}{\rho \sum_k w_{ik} + 1 - \rho},\ \frac{\sigma_R^2}{\rho \sum_k w_{ik} + 1 - \rho}\right) \quad \text{for } i = 1, \ldots, 2148 \text{ areas.} \tag{3.10}$$

That is, the expected random effects are weighted averages of the random effects of neighbouring areas, $R_{-1} = \{R_1, R_2, \ldots, R_{i-1}, R_{i+1}, \ldots, R_N\}$ with weight $\rho$ and have a global mean of 0 with weight $(1 - \rho)$. If the spatial dependence parameter, $\rho$, equals 1, the Leroux prior is the same as the intrinsic CAR prior [30], and if $\rho$ equals 0, the areas are spatially independent and no spatial smoothing occurs. The term $W_{ik}$ is the $(i, k)$th element of $N \times N$ spatial adjacency matrix having value 1 if areas $i$ and $k$ are considered to be neighbours and 0 otherwise. Neighbours are generally defined by shared boundaries, although islands are assigned neighbours predominately based on the closest mainland access points. An inverse Gamma prior is specified for the variance $\sigma_R^2$ and a uniform (0, 1) prior is chosen for the spatial-dependent parameter, $\rho$. For further details, see https://atlas.cancer.org.au/methodology/model-for-cancer-diagnosis/.

In the Leroux model applied to observed incidence data (equations (3.8)–(3.10)), the regions, major cities, regional and remote were used as covariates and a spatial random effects term ($R_i$) was included to account for spatial autocorrelation. In the proposed meta-analysis model, the regions are added in a hierarchy of the model (see equations (3.1)–(3.7)) and no spatial term was included explicitly since this was already in the reported ACA estimates.

To compare the results of the proposed meta-analysis model and the Leroux model fitted to the observed (real) areal data of cancer incidence, the relative difference between the posterior estimates

**Table 1.** Summary of posterior samples for lung cancer.

| | males | | | females | | |
|---|---|---|---|---|---|---|
| | mean | 0.025th quantile | 0.975th quantile | mean | 0.025th quantile | 0.975th quantile |
| deviance | −1578.190 | −1698.225 | −1455.656 | −2042.063 | −2153.995 | −1928.675 |
| $\mu_0$ | 0.049 | −0.362 | 0.459 | 0.013 | −0.388 | 0.415 |
| $\tau_1$ | 28.618 | 25.518 | 31.921 | 39.242 | 35.668 | 42.984 |
| $\tau_2$ | 31.828 | 28.249 | 35.541 | 35.810 | 32.125 | 39.641 |
| $\tau_3$ | 14.205 | 10.495 | 35.541 | 14.965 | 11.151 | 19.058 |
| $\tau_0$ | 11.943 | 2.129 | 18.266 | 12.387 | 2.200 | 26.940 |
| $\theta_1$ | −0.058 | −0.074 | −0.042 | −0.030 | −0.044 | −0.016 |
| $\theta_2$ | 0.034 | 0.016 | 0.053 | 0.006 | −0.012 | 0.023 |
| $\theta_3$ | 0.172 | 0.100 | 0.245 | 0.063 | −0.006 | 0.133 |

of region-specific means of the meta-analysis model (denoted by $e^{\theta_j}$) and the unit record model (denoted by $e^{\beta_j}$), was calculated as

$$\text{relative difference } (\%) = \frac{\exp(\theta_j) - \exp(\beta_j)}{\exp(\theta_j)} \times 100. \tag{3.11}$$

The posterior distribution of ranks of the region-specific mean SIRs (for major cities, regional and remote areas) are also calculated for both models.

# 4. Results

The proposed model was applied to analyse the outputs of each of the 20 cancers included in the ACA for both males and females, where applicable. In §4.1, the detailed results obtained by fitting the proposed model for two cancers, lung cancer and thyroid cancers for males and females are reported. Following this, the findings for all the cancers are summarized in §4.2. This is followed in §4.3 by a comparison of the outputs of meta-analyses with those found by analysing the raw incidence data in order to validate the proposed modelling approach.

## 4.1. Extended results for two cancers

### 4.1.1. Lung cancer

The summary estimates from the posterior samples for the parameters of interest for lung cancer (males and females), obtained from fitting the proposed meta-analysis model by running MCMC chains (§3.1.2), are given in table 1. The table shows that the overall fitted mean of log(SIR), calculated by $\mu_0$, is 0.049 for males (95% CI: −0.362, 0.459), i.e. the overall SIR of 1.05 (95% CI: 0.696,1.582). The corresponding estimates for females are an overall mean log(SIR) of 0.013 (95% CI: −0.388, 0.415), i.e. an overall SIR of 1.01 (95% CI: 0.674,1.514). The estimates of $\theta_1$, $\theta_2$ and $\theta_3$ are the region-specific means of the log(SIR) in major cities, regional and remote areas, respectively. From the summary results, we can see that the standardized incidence of lung cancer is relatively higher than the Australian average in remote areas and regional areas, and lower than the Australian average in major cities.

For a more detailed comparative analysis of geographical disparities in lung cancer incidence, the region-specific means, $\theta_j$, were ranked in descending order (region with highest mean log(SIR) is ranked 1) in each iteration of the MCMC chains generated for fitting the proposed model. The summaries of the posterior distribution of ranks of each region and the probability distribution of ranks per region are shown in tables 2 and 3.

From tables 2 and 3 of the ranks of the area-specific means, it is clear that areas within major cities are more likely to have lower incidence of lung cancer and remote areas are most likely to have higher

**Table 2.** Summary of posterior distribution of ranks for region-specific mean for lung cancer incidence.

| | median rank | | 95% credible interval | |
| --- | --- | --- | --- | --- |
| remoteness category | male | female | male | female |
| major cities | 3 | 3 | (2,3) | (2,3) |
| inner and outer regional | 2 | 2 | (2,3) | (2,3) |
| remote | 1 | 1 | (1,1) | (1,1) |

**Table 3.** Probability distribution of region-specific ranks for lung cancer.

| | probabilities | | | | | |
| --- | --- | --- | --- | --- | --- | --- |
| | major cities | | regional | | remote | |
| rank | male | female | male | female | male | female |
| 1 | 0.000 | 0.000 | 0.000 | 0.021 | 1.000 | 0.979 |
| 2 | 0.000 | 0.001 | 0.999 | 0.979 | 0.000 | 0.021 |
| 3 | 1.000 | 0.999 | 0.000 | 0.021 | 0.000 | 0.001 |

**Table 4.** Pairwise comparison of region specific mean for lung cancer.

| | probability | |
| --- | --- | --- |
| pairs | male | female |
| major cities > regional | 0 | 0.001 |
| major cities > remote | 0.001 | 0.006 |
| regional > remote | 0.172 | 0.058 |

**Table 5.** Posterior mean differences (pairwise) with 95% credible intervals for lung cancer.

| | males | | females | |
| --- | --- | --- | --- | --- |
| | posterior | 95% CI | posterior | 95% CI |
| pairs | mean difference | | mean difference | |
| major cities–regional | −0.092 | (−0.117, −0.068) | −0.036 | (−0.057, −0.013) |
| regional–remote | −0.138 | (−0.212, −0.064) | −0.057 | (−0.129, 0.015) |
| major cities–remote | −0.230 | (−0.157, −0.303) | −0.093 | (−0.022, −0.164) |

incidence of lung cancer for both sexes. This is supported by the pairwise comparisons given in table 4, the pairwise posterior mean differences of log(SIR), shown in table 5 and the posterior relative risks of SIRs for regional and remote areas with reference to major cities shown in table 6.

This is also well established in the literature that lung cancer is known as a low socio-economic status (SES) cancer, having higher risk in the areas with low SES neighbourhood [54,55]. In Australia, lung cancer risk is more in remote areas in comparison to major cities and regional areas [56]. So the output of the proposed meta-analysis models for lung cancer by remoteness categories are supported by the existing literature as well.

Figures 1 and 2 show the SIRs of lung cancer (for males and females) from the ACA and the fitted Bayesian hierarchical meta-analysis models. There is larger variability in the SIRs from the atlas

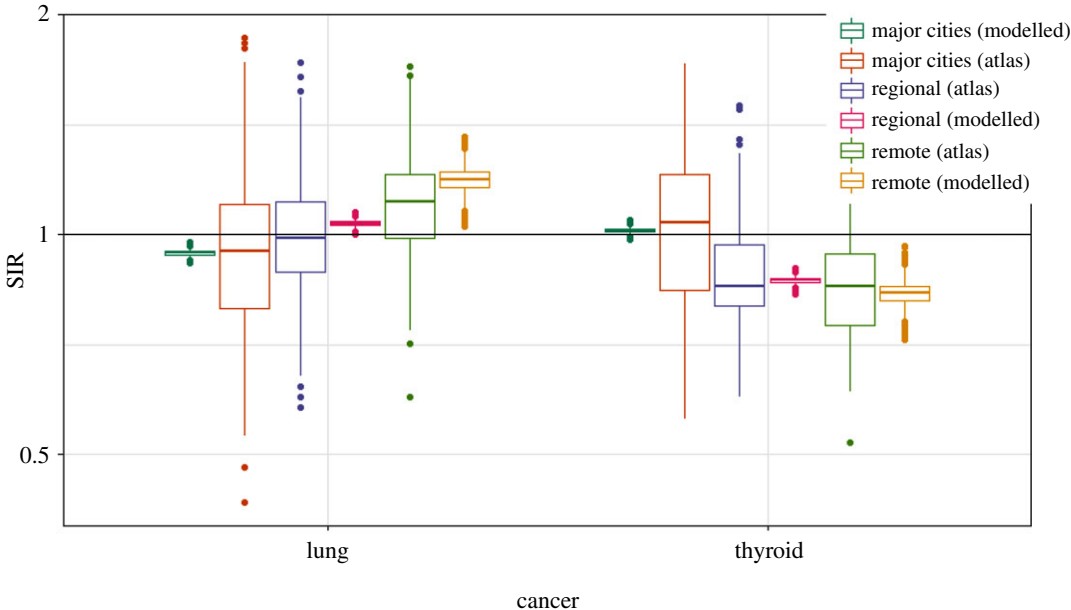

**Figure 1.** Boxplots of SIRs per region from ACA and fitted model for lung and thyroid cancers (males).

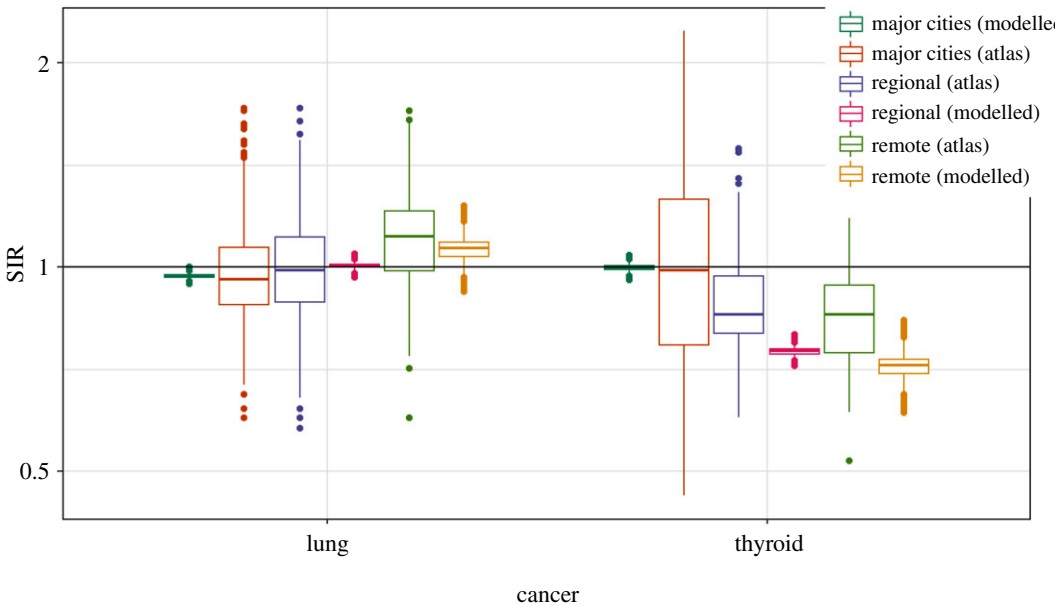

**Figure 2.** Boxplots of SIRs per region from ACA and fitted model for lung and thyroid cancers (females).

**Table 6.** Posterior relative risks for lung cancer in regional and remote Australia (baseline: major cities).

| remoteness category | relative risk (95% credible interval) | | | |
|---|---|---|---|---|
| | males | | females | |
| regional | 1.096 | (1.093, 1.099) | 1.037 | (1.032, 1.040) |
| remote | 1.259 | (1.190, 1.333) | 1.097 | (1.083, 1.162) |

compared with those of the fitted model, since the effect of remoteness has been removed from the estimates in our fitted model. The atlas estimates are obtained grouping all small-area estimates by remoteness categories without considering remoteness in the model, whereas in the proposed model, the posterior SIRs are the result of a model where remoteness categories are included as a covariate to obtain region-specific SIRs for major cities, regional and remote areas.

**Table 7.** Summary of posterior samples for thyroid cancer.

| | males | | | females | | |
|---|---|---|---|---|---|---|
| | mean | 0.025th quantile | 0.975th quantile | mean | 0.025th quantile | 0.975th quantile |
| deviance | −1617.290 | −1733.616 | −1498.474 | −940.021 | −1077.563 | −800.930 |
| $\mu_0$ | −0.104 | −0.514 | 0.307 | −0.206 | −0.636 | 0.221 |
| $\tau_1$ | 28.658 | 25.660 | 31.818 | 14.801 | 13.111 | 16.652 |
| $\tau_2$ | 33.874 | 30.209 | 37.731 | 18.963 | 16.187 | 22.041 |
| $\tau_3$ | 16.103 | 12.252 | 20.306 | 14.793 | 11.035 | 18.893 |
| $\tau_0$ | 12.088 | 2.112 | 26.621 | 11.248 | 1.849 | 24.984 |
| $\theta_1$ | 0.012 | −0.003 | 0.028 | −0.001 | −0.020 | 0.019 |
| $\theta_2$ | −0.143 | −0.162 | −0.125 | −0.285 | −0.309 | −0.261 |
| $\theta_3$ | −0.183 | −0.250 | −0.117 | −0.334 | −0.405 | −0.263 |

**Table 8.** Summary of posterior distribution of ranks for region-specific means for thyroid cancer incidence.

| | median rank | | 95% credible interval | |
|---|---|---|---|---|
| remoteness category | male | female | male | female |
| major cities | 1 | 1 | (1,3) | (1,3) |
| inner and outer regional | 2 | 2 | (1,3) | (1,3) |
| remote | 3 | 3 | (1,3) | (1,3) |

**Table 9.** Probability distribution of region-specific ranks for thyroid cancer.

| | probabilities | | | | | |
|---|---|---|---|---|---|---|
| | major cities | | regional | | remote | |
| rank | male | female | male | female | male | female |
| 1 | 1.000 | 0.999 | 0.000 | 0.058 | 0.000 | 0.000 |
| 2 | 0.000 | 0.001 | 0.878 | 0.942 | 0.122 | 0.102 |
| 3 | 0.000 | 0.000 | 0.122 | 0.000 | 0.898 | 0.898 |

### 4.1.2. Thyroid cancer

The summary estimates from the posterior samples for the parameters of interest for thyroid cancer (males and females) are given in table 7. The table shows that the overall fitted mean log(SIR) is −0.104 for males (95% CI: −0.514, 0.307), i.e. the overall SIR of 0.901 (95% CI: 0.598, 1.359). The corresponding estimates for females are −0.206 (95% CI: −0.636, 0.221), i.e. an overall SIR of 0.814 (95% CI: 0.529, 1.247). The estimates of $\theta_1$, $\theta_2$ and $\theta_3$ are the specific means of the log(SIR) in major cities, regional and remote areas, respectively. From the summary results, we can see that the standardized incidence of thyroid cancer is relatively higher than the Australian average in major cities and lower than the Australian average in regional and remote areas.

The summaries of the posterior distribution of ranks of each region and the probability distribution of ranks per region for thyroid cancer are shown in tables 8 and 9.

From tables 8 and 9, it is apparent that thyroid cancer is most likely to have higher incidence in the areas within major cities for both sexes, which is supported by the pairwise comparisons given in table 10, pairwise

**Table 10.** Pairwise comparison of region-specific mean for thyroid cancer.

| | probability | |
| --- | --- | --- |
| pairs | male | female |
| major cities > regional | 1.000 | 1.000 |
| major cities > remote | 1.000 | 1.000 |
| regional > remote | 0.878 | 0.898 |

**Table 11.** Posterior Mean Differences (pairwise) with 95% credible intervals for thyroid cancer.

| | males | | females | |
| --- | --- | --- | --- | --- |
| | posterior | 95% CI | posterior | 95% CI |
| pairs | mean difference | | mean difference | |
| major cities–regional | 0.155 | (0.131, 0.180) | 0.285 | (0.254, 0.315) |
| regional–remote | 0.040 | (−0.028, 0.110) | 0.049 | (−0.026, 0.125) |
| major cities–remote | 0.196 | (0.264, 0.128) | 0.333 | (0.407, 0.260) |

**Table 12.** Posterior relative risks for thyroid cancer in regional and remote Australia (baseline: major cities).

| | relative risk (95% credible interval) | | | |
| --- | --- | --- | --- | --- |
| remoteness category | males | | females | |
| regional | 0.856 | (0.8536, 0.8585) | 0.753 | (0.7490, 0.7557) |
| remote | 0.823 | (0.7812, 0.8647) | 0.717 | (0.6801, 0.7544) |

posterior mean differences of log(SIR), shown in table 11, and the posterior relative risks of SIRs for regional and remote areas with reference to major cities in table 12. Figures 1 and 2 also depicts the SIRs of thyroid cancer (for males and females) from the ACA and the fitted Bayesian hierarchical meta-analysis models.

Regarding the mismatches in the probabilities and posterior mean differences, the width of credible intervals constructed for the posterior mean differences obviously reflects the amount of variation in these estimates. While calculating the probabilities, we calculated the proportion of times the fitted mean for major cities is larger than the corresponding means for regional and remote areas, and so on. We did not consider the magnitude of the differences or the uncertainty of the obtained probabilities. We inferred differences between regions based on the credible intervals for the posterior mean differences.

## 4.2. Summary of outputs for Bayesian meta-analysis model

A summary of outputs for all the 20 cancers included in ACA is reported in this section. Table 13 shows the pairwise probabilities of major cities having higher SIR than regional and remote regions, as well as the probabilities of regional areas exceeding remote regions with respect to SIR, for each of the 20 cancers (by sex where applicable). For instance, on average, the SIR in major cities is almost certainly larger than the SIR in regional and remote areas for eight cancer/sex groups (for males: non-Hodgkin lymphoma, stomach and for females: breast, myeloma, non-Hodgkin lymphoma, ovarian, stomach, thyroid). Conversely, on average, major cities are not likely or substantially less likely to have higher SIRs than regional and remote areas for 10 cancer sex group (for males: bowel, head and neck, lung, oesophageal and for females: bowel, cervical, head and neck, lung, melanoma, oesophageal). On the contrary, for leukaemia (females), the probability that the SIR is larger in major cities than regional areas is 0.476 which indicates little difference between the SIR in major cities and regional areas, on average. Similar interpretation can be made for pancreatic cancer (for males: $P$(major cities > remote) =

**Table 13.** Pairwise comparison and average ranks of regions for all cancers (males and females).

| cancer | sex | probabilities | | | ranks | | |
| | | major cities > regional | major cities > remote | regional > remote | major cities | regional | remote |
|---|---|---|---|---|---|---|---|
| all cancers | male | 0.00 | 0.992 | 0.999 | 2 | 1 | 3 |
| | female | 0.012 | 0.943 | 0.985 | 2 | 1 | 3 |
| bowel cancer | male | 0.000 | 0.145 | 0.974 | 3 | 1 | 2 |
| | female | 0.000 | 0.468 | 0.994 | 3 | 1 | 2 |
| brain cancer | male | 0.578 | 0.892 | 0.878 | 1 | 2 | 3 |
| | female | 0.716 | 0.826 | 0.777 | 1 | 2 | 3 |
| breast cancer | female | 1.000 | 0.999 | 0.992 | 1 | 2 | 3 |
| cervical cancer | female | 0.000 | 0.000 | 0.000 | 3 | 2 | 1 |
| head and neck cancer | male | 0.000 | 0.000 | 0.000 | 3 | 2 | 1 |
| | female | 0.000 | 0.000 | 0.000 | 3 | 2 | 1 |
| kidney cancer | male | 0.605 | 0.995 | 0.994 | 1 | 2 | 3 |
| | female | 0.007 | 0.784 | 0.935 | 2 | 1 | 3 |
| leukaemia | male | 0.257 | 0.735 | 0.788 | 2 | 1 | 3 |
| | female | 0.936 | 0.476 | 0.302 | 2 | 3 | 1 |
| liver cancer | male | 1.000 | 0.456 | 0.000 | 2 | 3 | 1 |
| | female | 1.000 | 0.129 | 0.000 | 2 | 3 | 1 |
| lung cancer | male | 0.000 | 0.000 | 0.0001 | 3 | 2 | 1 |
| | female | 0.001 | 0.006 | 0.061 | 3 | 2 | 1 |
| melanoma | male | 0.000 | 1.000 | 1.000 | 2 | 1 | 3 |
| | female | 0.000 | 0.432 | 1.000 | 3 | 1 | 2 |
| myeloma | male | 0.652 | 0.955 | 0.942 | 1 | 2 | 3 |
| | female | 0.998 | 0.992 | 0.941 | 1 | 2 | 3 |
| non-Hodgkin lymphoma | male | 1.000 | 1.000 | 0.999 | 1 | 2 | 3 |
| | female | 1.000 | 0.999 | 0.937 | 1 | 2 | 3 |
| oesophageal cancer | male | 0.000 | 0.000 | 0.0002 | 3 | 2 | 1 |
| | female | 0.000 | 0.000 | 0.0001 | 3 | 2 | 1 |
| ovarian cancer | female | 0.996 | 0.955 | 0.816 | 1 | 2 | 3 |
| pancreatic cancer | male | 0.766 | 0.573 | 0.487 | 1 | 2 | 3 |
| | female | 0.587 | 0.725 | 0.698 | 1 | 2 | 3 |
| prostate cancer | male | 0.440 | 1.000 | 1.000 | 2 | 1 | 3 |
| stomach cancer | male | 1.000 | 1.000 | 0.953 | 1 | 2 | 3 |
| | female | 1.000 | 0.999 | 0.904 | 1 | 2 | 3 |
| thyroid cancer | male | 1.000 | 1.000 | 0.878 | 1 | 2 | 3 |
| | female | 1.000 | 1.000 | 0.898 | 1 | 2 | 3 |
| uterine cancer | female | 0.769 | 0.069 | 0.046 | 2 | 3 | 1 |

0.573, $P$(regional > remote) = 0.487; for females: $P$(major cities > regional) = 0.587), bowel cancer (for females: $P$(major cities > regional = 0.587) and kidney cancer (for males: $P$(major cities > regional) = 0.605). So a probability close to 0.5 does not indicate substantial difference between the pair under comparison. However, the ranking of the three regions is still able to show a pattern, so in the mentioned cases the pattern is suggestive, not substantial. In addition to the ranks and probability, table 14 shows the relative risks for each cancer by sex. The relative risks shown in table 14 provide

**Table 14.** Relative risks for cancers by remoteness categories and sex (reference: major cities).

| cancers | relative risk (95% credible intervals) | | | | | | | |
| --- | --- | --- | --- | --- | --- | --- | --- | --- |
| | males | | | | females | | | |
| | regional | | remote | | regional | | remote | |
| all | 1.038 | (1.036, 1.043) | 0.931 | (0.897, 0.978) | 1.016 | (1.013, 1.018) | 0.961 | (0.922, 1.001) |
| bowel | 1.088 | (1.086, 1.093) | 1.033 | (0.985, 1.079) | 1.076 | (1.072, 1.079) | 1.002 | (0.958, 1.048) |
| brain | 0.989 | (0.995, 1.000) | 0.957 | (0.921, 1.011) | 0.995 | (0.992, 0.999) | 0.974 | (0.931, 1.019) |
| breast | | | | | 0.969 | (0.966, 0.972) | 0.908 | (0.869, 0.947) |
| cervical | | | | | 1.054 | (1.048, 1.059) | 1.243 | (1.173, 1.318) |
| head and neck | 1.469 | (1.270, 1.281) | 1.677 | (1.585, 1.775) | 1.116 | (1.111, 1.121) | 1.280 | (1.213, 1.349) |
| kidney | 0.998 | (0.994, 1.000) | 0.923 | (0.879, 0.969) | 1.026 | (1.022, 1.031) | 0.974 | (0.924, 1.027) |
| leukaemia | 1.006 | (1.001, 1.009) | 0.982 | (0.938, 1.028) | 0.985 | (0.981, 0.989) | 1.002 | (0.953, 1.054) |
| liver | 0.824 | (0.821, 0.829) | 1.005 | (0.936, 1.083) | 0.840 | (0.836, 0.844) | 1.054 | (0.979, 1.134) |
| lung | 1.096 | (1.093, 1.099) | 1.259 | (1.190, 1.333) | 1.037 | (1.032, 1.040) | 1.097 | (1.083, 1.162) |
| melanoma | 1.125 | (1.128, 1.126) | 0.834 | (0.789, 0.875) | 1.241 | (1.246, 1.242) | 1.021 | (0.957, 1.062) |
| myeloma | 0.996 | (0.992, 1.000) | 0.945 | (0.897, 0.996) | 0.973 | (0.969, 0.977) | 0.926 | (0.882, 0.973) |
| nH lymphoma | 0.959 | (0.956, 0.963) | 0.868 | (0.830, 0.908) | 0.949 | (0.946, 0.952) | 0.909 | (0.870, 0.950) |
| oesophageal | 1.239 | (1.235, 1.245) | 1.403 | (1.332, 1.480) | 1.239 | (1.235, 1.246) | 1.404 | (1.331, 1.481) |
| ovarian | | | | | 0.979 | (0.976, 0.982) | 0.955 | (0.913, 0.997) |
| pancreatic | 0.993 | (0.990, 0.997) | 0.995 | (0.949, 1.042) | 0.998 | (0.994, 1.000) | 0.984 | (0.940, 1.029) |
| prostate | 1.001 | (0.998, 1.005) | 0.790 | (0.747, 0.835) | | | | |
| stomach | 0.934 | (0.931, 0.938) | 0.882 | (0.838, 0.929) | 0.906 | ( 0.903, 0.909) | 0.867 | (0.824, 0.912) |
| thyroid | 0.856 | (0.854, 0.859) | 0.823 | (0.781, 0.865) | 0.754 | (0.749, 0.756) | 0.717 | (0.680, 0.754) |
| uterine | | | | | 0.993 | (0.989, 0.997) | 1.048 | (0.997, 1.101) |

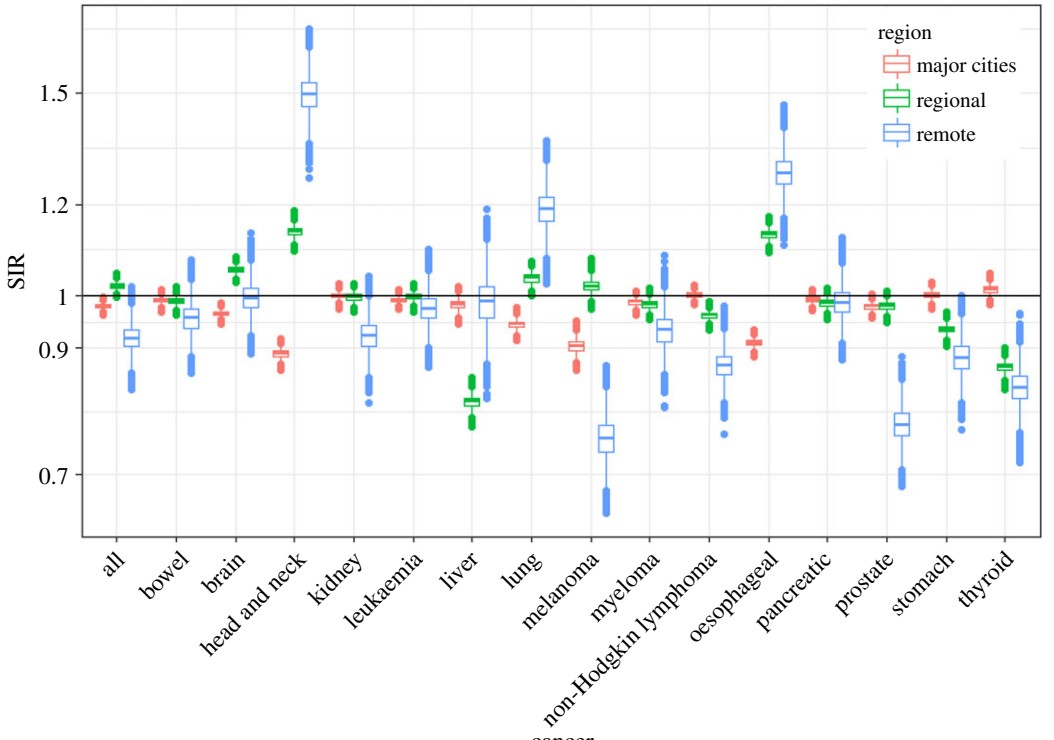

**Figure 3.** Box plots of Bayesian meta-analysis fitted SIR for males by region and cancer type.

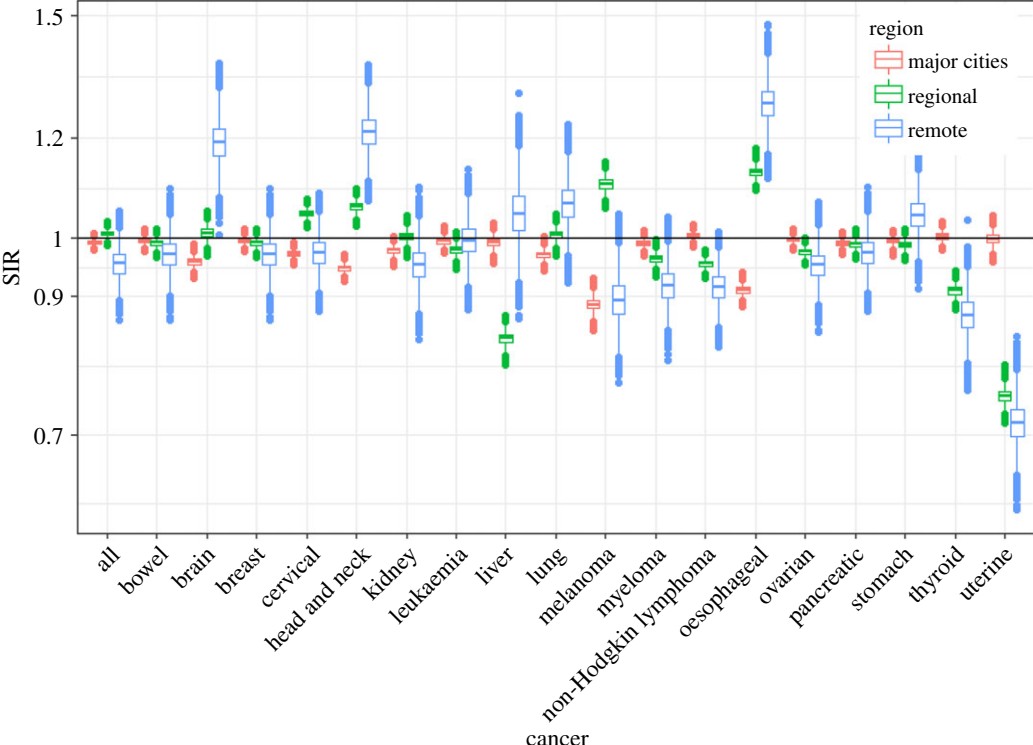

**Figure 4.** Box plots of Bayesian meta-analysis fitted SIR for females by region and cancer type.

the magnitude increase or decrease in the SIRs of regional and remote areas in Australia compared with those of major cities, on the average.

Figures 3 and 4 show the fitted values of SIRs obtained from the proposed meta-analysis model for major cities, regional and remote areas for each cancer for males and females, respectively; the horizontal line represents the Australian average. These figures show, at a glance, the large variation in the SIRs by remoteness categories. A comparison of these fitted values to the SIRs from the atlas is shown in figures 5

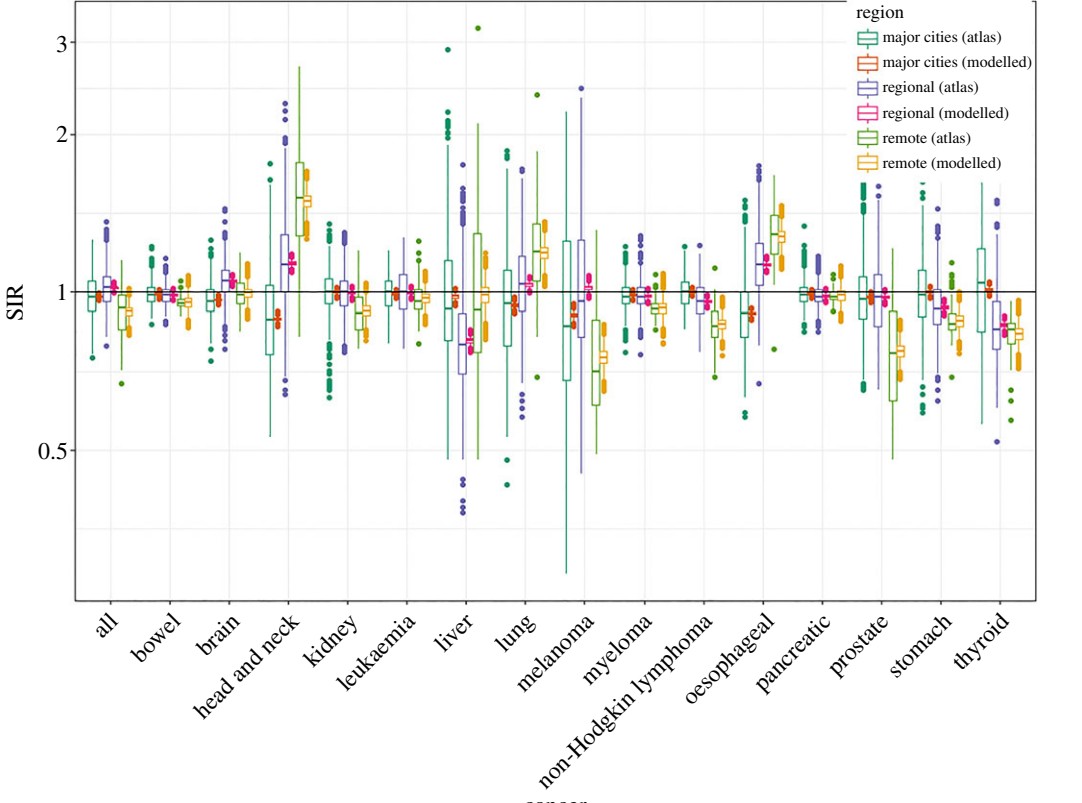

**Figure 5.** Boxplots of SIRs per regions from ACA and fitted model for all cancers for males.

and 6. The comparison confirms that the two sets of figures show the same patterns of cancer incidence by remoteness of the regions to that present in the atlas data, i.e. for each of the cancers, the region with highest or lowest SIRs are the same despite having a different spread of the values.

## 4.3. Validation of outputs of Bayesian meta-analysis model

In this section, the region-specific estimates obtained from the proposed Bayesian hierarchical meta-analysis model on the ecological data (equations (3.1)–(3.7)) and the Leroux model using the observed areal incidence data with remoteness categories as covariates (equations (3.8)–(3.10)) are compared.

The posterior distribution of ranks of the region-specific mean SIRs (for major cities, regional and remote areas) for each cancer revealed consistently the same patterns. For instance, according to both models, SIR of brain cancer is most likely to have higher values in major cities compared with regional and remote areas, melanoma is most likely to have a higher SIR in regional areas compared with major cities and remote areas (table 13).

The difference in posterior estimates of mean SIR for each cancer, for males and females, between the output of unit record model and Bayesian hierarchical meta-analysis model are displayed in figures 7 and 8 respectively. Table 15 shows the relative difference between the posterior estimates of region-specific means of the meta-analysis model and the unit record model.

The relative differences between the posterior means for major cities and regional areas are small, with the relative differences being less than 5% for most of the cancers, and less than 10% for only four cancers (table 15). However, the relative differences between the posterior mean estimates for remote areas are much larger. This is because of the smaller number of remote areas and the greater variability between the estimated SIRs in this category. See appendix A.2. for further exposition.

## 5. Model extensions

The proposed Bayesian hierarchical meta-analysis models have been applied to ACA data to identify relationship between remoteness categories and incidence of 20 different cancer types by sex. The

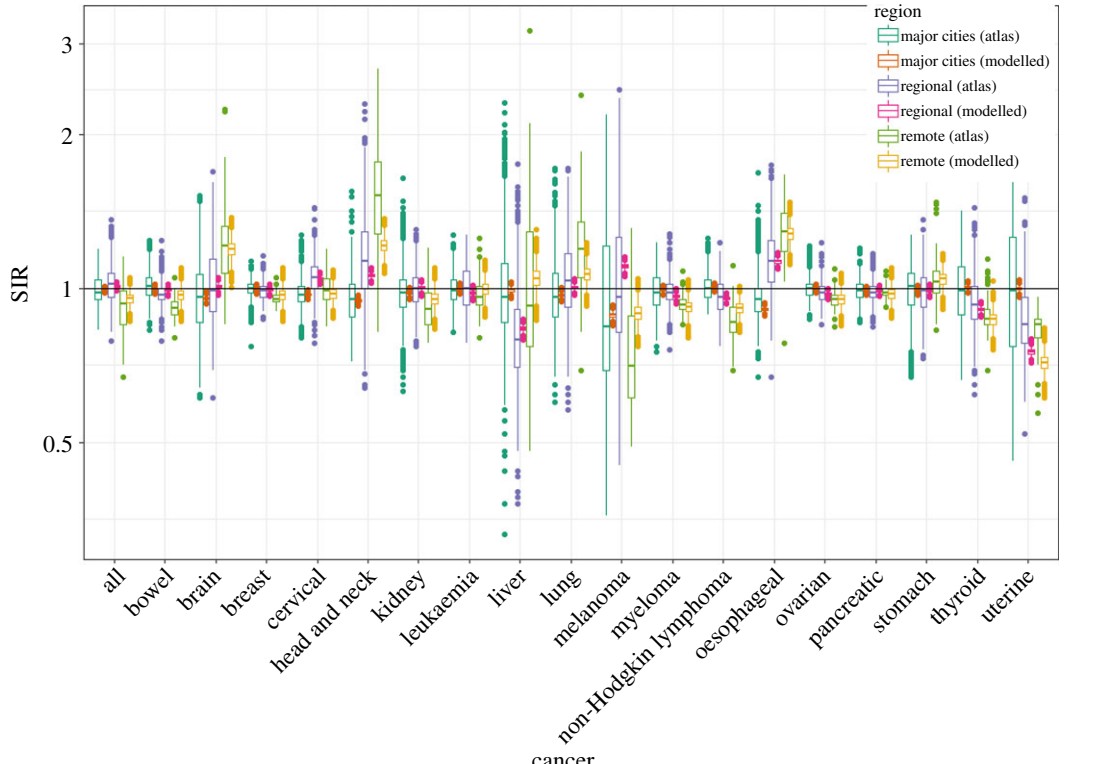

**Figure 6.** Boxplots of SIRs per regions from ACA and fitted model for all cancers for females.

proposed model could be extended in a number of ways. Two examples of these extensions are the inclusion of a spatial component and the inclusion of continuous covariates.

## 5.1. Inclusion of a spatial component

In the present study, we have modelled spatially smoothed estimates of log(SIR) from ACA. Since the estimated SIRs were already results of a Bayesian spatial model (with a Leroux prior for the spatial term), we did not use any spatial component in our proposed hierarchical Bayesian meta-analysis model. Further investigation of the residuals of the fitted models for 20 different cancer types by sex resulted in Moran's $I$ values ranging from 0.15380 to 0.89485 with corresponding standard deviations from 0.01447 to 0.01450 and all $p$-values less than 0.0001 (see appendix A.3, table 22). To improve the model performance, a spatial term could be added to the proposed meta-analysis model and a spatial prior could be chosen for modelling. This would be a straightforward extension to the specified model (equations (3.1)–(3.7)); instead of equation (3.1) in the present model, we can adopt the following:

$$Y_{i[j]} = \mu_{i[j]} + \psi_{i[j]} + \epsilon_{i[j]}, \tag{5.1}$$

where, $\epsilon_{i[j]} \sim N(0, \sigma^2_{i[j]})$ is an unstructured error component and $\psi_{i[j]}$ is a spatial random effect having a spatial prior. Any suitable spatial prior could be chosen to model the spatial component [30,31]. For illustration, the proposed Bayesian meta-analysis model with spatial component is fitted for two cancer types, pancreatic and liver cancer (males), which have Moran's $I$ values 0.15380 and 0.89485, respectively (table 22). The updated model codes adding a spatial component are available in appendix A.4.

### 5.1.1. Meta-analysis models with spatial component for liver and pancreatic cancer (males)

The Bayesian hierarchical meta-analysis model with spatial component introduced at first step as shown by equation (5.1) are fitted in R using *R2WinBUGS* package. We fitted the model for liver and pancreatic cancer (males). The results are summarized in this section.

The posterior estimates of the model parameters and the corresponding 95% credible intervals for the two cancers are presented in table 16. The posterior estimates of region-specific means $\theta_j$, $j = 1, 2, 3$ for each of the three regions (major cities, regional and remote areas) are ranked and the ranks are summarized in

**Table 15.** relative differences in posterior means for SIRs by region.

| cancers | relative differences (%) | | | | | |
| --- | --- | --- | --- | --- | --- | --- |
| | males | | | females | | |
| | major cities | regional | remote | major cities | regional | remote |
| all | −1.15 | 1.28 | −1.66 | −0.83 | 1.71 | −1.16 |
| bowel | −0.52 | −2.77 | 3.49 | −0.92 | −1.13 | 4.53 |
| brain | −0.63 | −1.05 | 27.64 | −0.19 | −0.14 | 16.02 |
| breast | | | | −0.09 | 0.79 | 9.11 |
| cervical | | | | −2.47 | 1.19 | −6.79 |
| head and neck | −6.92 | 2.85 | 8.15 | 0.66 | −7.79 | −27.16 |
| kidney | −1.66 | 4.46 | 8.30 | 0.27 | −3.17 | 3.99 |
| leukaemia | 0.27 | −1.07 | −0.18 | −0.39 | 1.36 | −3.15 |
| liver | 0.72 | −3.65 | 3.02 | −2.67 | 9.03 | −25.35 |
| lung | −1.38 | −1.15 | 3.32 | −2.10 | 3.60 | −19.26 |
| melanoma | 0.202 | −7.64 | −15.20 | −0.89 | −8.28 | −16.64 |
| myeloma | −1.24 | 0.59 | 31.41 | −1.02 | 1.73 | 25.37 |
| nH lymphoma | 0.24 | −1.03 | 14.41 | −1.16 | 3.44 | 14.47 |
| oesophageal | −0.84 | −8.01 | −12.66 | −2.51 | 1.65 | 6.09 |
| ovarian | | | | −0.81 | 1.59 | 15.63 |
| pancreatic | −1.46 | 3.45 | 1.90 | −0.60 | 1.05 | 1.74 |
| prostate | 1.21 | −2.92 | −15.41 | | | |
| stomach | 0.10 | 0.49 | 13.50 | 0.00 | 0.00 | 0.00 |
| thyroid | 4.38 | −9.75 | 4.34 | 0.00 | 0.00 | 0.00 |
| uterine | | | | 1.45 | −3.17 | −4.48 |

table 17. The ranks show that major cities have highest incidence of liver and pancreatic cancer. The median ranks are the same in terms of identifying the region with lowest incidence compared with those obtained by the model without the spatial component (table 13). However, for the region with highest incidence, the 95% credible intervals for both models agree in spite of differences in median ranks, as observed from the credible intervals of the ranks. The probabilities of major cities having larger incidence than regional and remote areas are shown in table 18, followed by the posterior mean differences with 95% credible intervals in table 19.

The posterior relative risks of liver and pancreatic cancer for the extended model (table 20) and the model without a spatial component (table 14) are also similar. Hence adding an additional spatial component to the Bayesian hierarchical meta-analysis model did not change the inference for the two cancers modelled here. In figures 9 and 10, we can observe the posterior mean SIRs for each region for both models. As expected, there are differences in the magnitude due to the additional spatial smoothing, but the observed pattern remains the same for both the cancers.

The relative differences of estimated log(SIR) in the three different remoteness regions using a meta-analysis model with a spatial component and the model using observed incidence data are shown in table 21. The relative differences of the model without spatial component (table 15) and the extended model with a spatial term are very close and they are all very small (less than 5%). However, the extended model could be fitted to cancers with larger relative differences such as myeloma, brain, prostate for males and liver, head and neck, ovarian for females to check whether adding a spatial component can reduce the relative difference from the estimates obtained using the observed data.

Moran's $I$ of the residuals after fitting the meta-analysis model with a spatial component for liver and pancreatic cancer (males) are 0.8857 (s.d. 0.1449, $p$-value < 0.0001) and 0.2051 (s.d. 0.0144, $p$-value < 0.0001), respectively, which are still statistically significant and very similar to what we

**Table 16.** Summary of posterior samples for liver and pancreatic cancer (males).

| | liver | | | pancreatic | | |
| | mean | 0.025th quantile | 0.975th quantile | mean | 0.025th quantile | 0.975th quantile |
| --- | --- | --- | --- | --- | --- | --- |
| deviance | −640.163 | −1032.000 | −341.800 | −3138.183 | −3274.000 | −3004.000 |
| $\mu_0$ | −0.090 | −0.497 | 0.319 | −0.011 | −0.409 | 0.381 |
| $\tau_1$ | 37.356 | 33.640 | 41.210 | 53.158 | 49.185 | 57.110 |
| $\tau_2$ | 32.383 | 28.670 | 36.420 | 44.294 | 40.395 | 48.250 |
| $\tau_3$ | 12.782 | 9.062 | 16.850 | 17.756 | 13.760 | 22.020 |
| $\tau_0$ | 12.296 | 2.135 | 27.200 | 12.472 | 2.236 | 27.261 |
| $\theta_1$ | −0.036 | −0.059 | −0.013 | −0.009 | −0.021 | 0.003 |
| $\theta_2$ | −0.171 | −0.202 | −0.140 | −0.014 | −0.030 | 0.002 |
| $\theta_3$ | −0.067 | −0.170 | 0.039 | −0.011 | −0.073 | 0.050 |

**Table 17.** Summary of posterior distribution of ranks for region specific mean for liver and pancreatic cancer (males).

| | liver | | pancreatic | |
| remoteness category | median ranks | 95% credible interval | median ranks | 95% credible interval |
| --- | --- | --- | --- | --- |
| major cities | 1 | (1,2) | 1 | (1,2) |
| inner and outer regional | 3 | (2,3) | 2 | (1,3) |
| remote | 2 | (1,2) | 3 | (1,3) |

**Table 18.** Pairwise comparison of region specific means.

| | probability | |
| pairs | liver | pancreatic |
| --- | --- | --- |
| major cities > regional | 1.000 | 0.691 |
| major cities > remote | 0.717 | 0.534 |
| regional > remote | 0.026 | 0.463 |

**Table 19.** Posterior mean differences (pairwise) with 95% credible intervals.

| | liver | | pancreatic | |
| pairs | posterior mean difference | 95% CI | posterior mean difference | 95% CI |
| --- | --- | --- | --- | --- |
| major cities–regional | 0.136 | (0.096, 0.175) | 0.006 | (−0.012, 0.024) |
| regional–remote | −0.104 | (−0.193, −0.016) | −0.003 | (−0.056, 0.049) |
| major cities–remote | 0.031 | (−0.064, 0.125) | 0.003 | (−0.051, 0.056) |

obtained by fitting the meta-analysis model without the spatial component (see appendix A.3, table 22). Hence adding another spatial component to the proposed meta-analysis model, which was modelling already spatially smoothed estimates of log(SIR) of each cancer, did not alleviate the spatial autocorrelation from the model residuals.

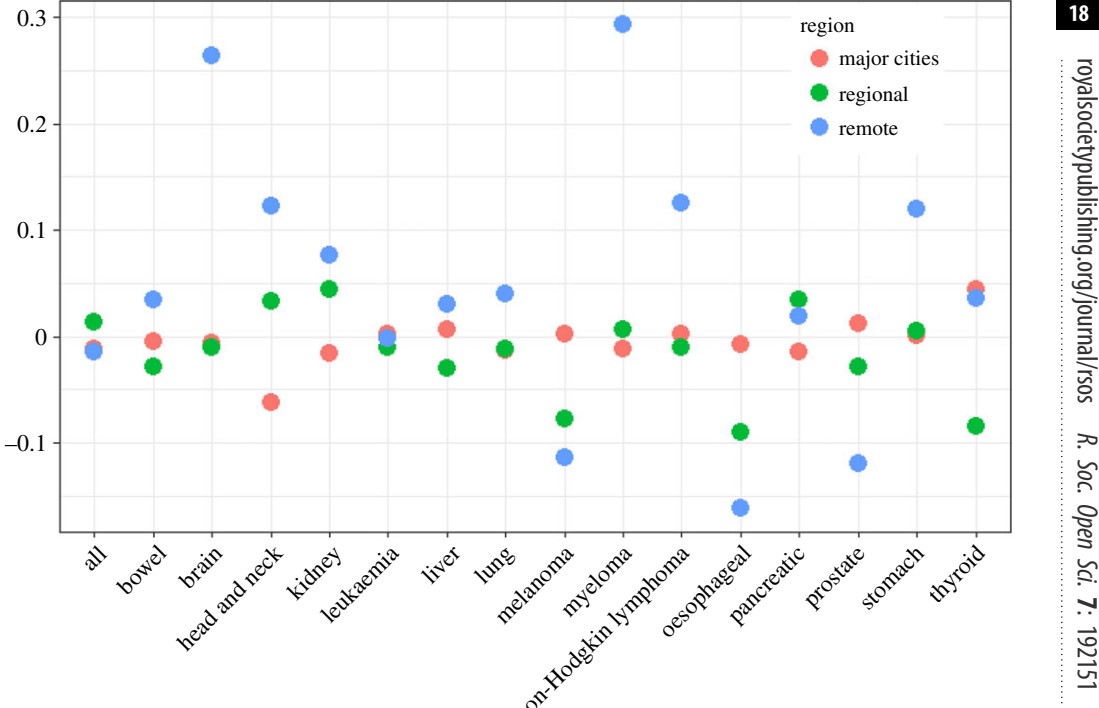

**Figure 7.** Actual differences in posterior mean SIR between the unit record model and meta-analysis model for males by region and cancer type.

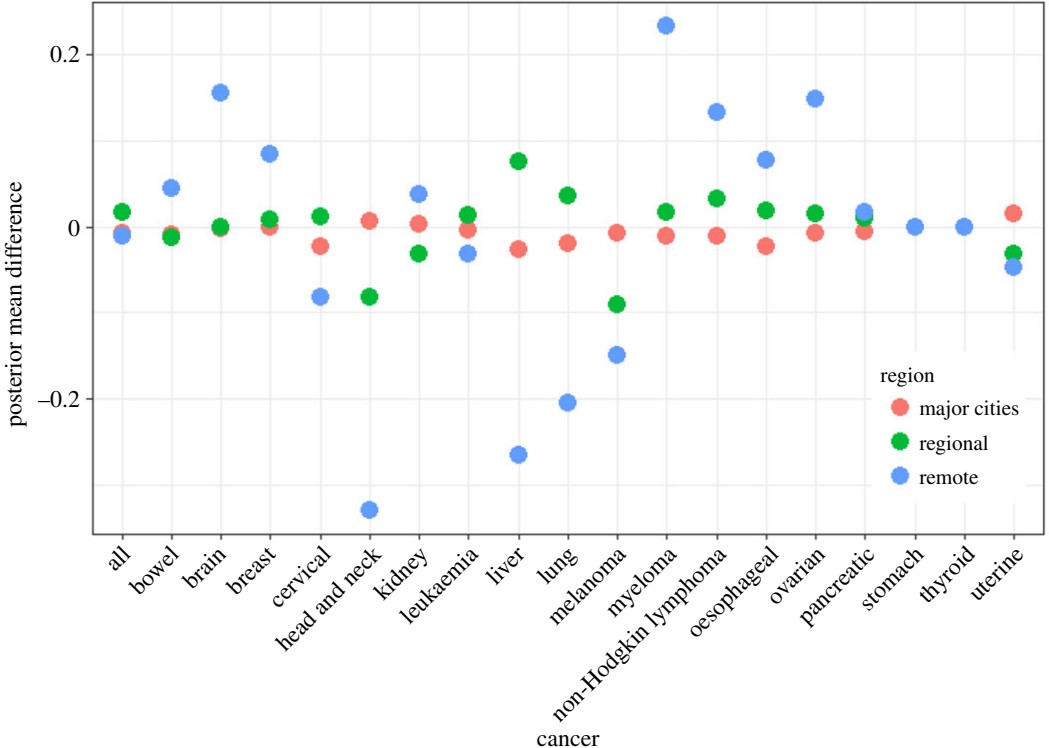

**Figure 8.** Actual differences in posterior mean SIR between the unit record model and meta-analysis model for females by region and cancer type.

From this further investigation, we recommend using the proposed Bayesian hierarchical meta-analysis model without the additional spatial component to model estimated cancer incidence data if the estimates were results of spatial models, for example, in case of ACA data. However, if the estimates are not obtained using a spatial random effects model, then including a spatial component in the meta-analysis

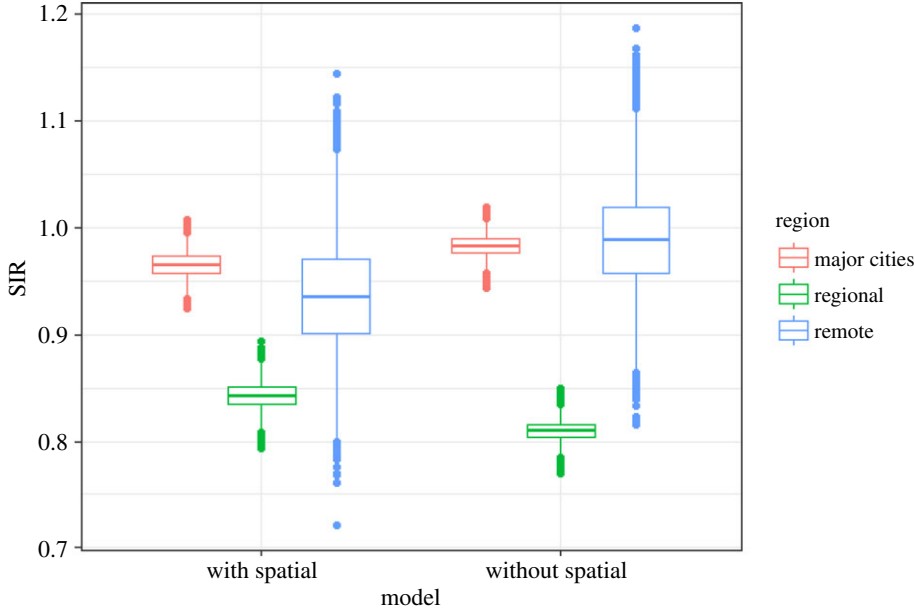

**Figure 9.** Posterior mean SIR of liver cancer (males) for meta-analysis models with and without spatial component.

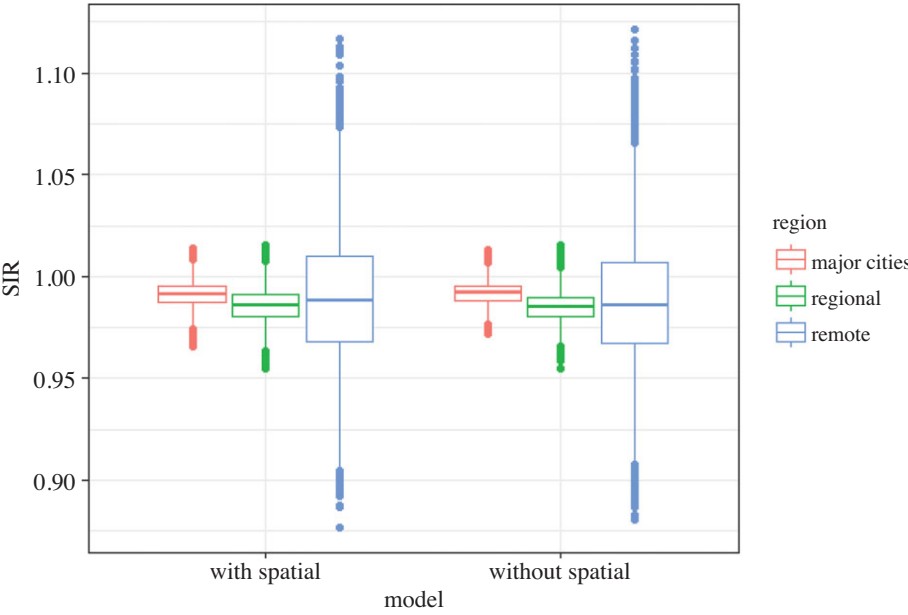

**Figure 10.** Posterior mean SIR of pancreatic cancer (males) for meta-analysis models with and without spatial component.

**Table 20.** Posterior relative risks for liver and pancreatic cancer for males regional and remote Australia (baseline: major cities).

| remoteness category | relative risk (95% credible interval) | | | |
|---|---|---|---|---|
| | liver | | pancreatic | |
| regional | 0.872 | (0.867, 0.879) | 0.995 | (0.991, 0.999) |
| remote | 0.966 | (0.889, 1.046) | 0.998 | (0.949, 1.048) |

**Table 21.** Relative differences in posterior means for SIRs by region.

| | relative differences (%) | | |
| --- | --- | --- | --- |
| cancers | major cities | regional | remote |
| liver | −1.07 | 1.58 | 4.06 |
| pancreatic | 1.48 | 2.16 | 1.78 |

model might be useful and indeed essential. It is also noted that these observations are based on this particular dataset; this motivates further research in order to make more general statements.

## 5.2. Adding continuous covariates

In the proposed Bayesian hierarchical meta-analysis model, only one covariate is included as a level of hierarchy, which is remoteness categories. The model can be modified to include continuous covariates if we replace the $\theta_j$ by $\theta$ in equation (3.2) as

$$\theta = X\beta, \tag{5.2}$$

where $X$ is a design matrix comprising continuous covariates and $\beta$ is the vector of corresponding regression coefficients. The resulting model equation can thus be rewritten as a meta regression model with normal priors on the regression coefficients [48].

## 6. Discussion

This study has proposed and illustrated a methodology for using information provided in the form of disease maps in atlases. The benefits of the proposed approach that has been demonstrated include the ability to gain further insights from the atlases without having access to the original data, which are sometimes unavailable due to privacy and other reasons. The utility of this approach has been illustrated by the substantive analysis of estimated point and interval estimates of SIRs available in the ACA.

The proposed methodology was applied to analyse the outputs from the ACA for 2148 small areas across Australia. This study aimed to determine whether a meta-analysis approach using modelled estimates was able to obtain similar inferences with those obtained using observed areal incidence data in identifying differences in cancer incidence by remoteness of an area. The output of the proposed meta-analysis model demonstrated patterns of cancer incidence by remoteness categories for most of the cancer types. It was observed that some cancers are more likely to occur in remote areas (head and neck cancers, liver cancer, lung cancer, oesophageal cancer for males and females and cervical cancer, uterine cancer for females), while some cancers are more likely to occur in regional areas (all cancers, bowel cancer, melanoma for both sexes, kidney cancer for females, leukaemia and prostate cancer for males) and some cancers are more likely to have greater incidence in major cities (brain, myeloma, non-Hodgkin lymphoma, pancreatic, stomach, thyroid cancer for both sexes, kidney cancer for males, leukaemia and ovarian cancer for females). Using the estimated SIRs reported in the ACA, the proposed model probabilistically identified these patterns of cancer incidence for different regions (major cities, regional and remote areas) in Australia. These probabilistic inferences could be useful to policy-makers by supplementing the information from the ACA.

The meta-analysis method used in our study is, in effect, calculating the average of already averaged SIR estimated for each SA2; this means that there will be some measurement error compared with average SIR estimates based on observed areal incidence data. However, given the slight differences in time periods, the general consistency between the remoteness patterns in cancer incidence we have reported in our study with those reported by the Australian Institute of Health and Welfare (AIHW) in the CIMAR (Cancer Incidence and Mortality Across Regions) books using observed areal incidence data [38] is encouraging. This is particularly so when we focus on our results that had very high (~1) or low (~0) probabilities of being higher or lower than other remoteness categories. Clearly, there is greater uncertainty when the probabilities are closer to 0.5, and so these patterns should be interpreted with much greater caution. As an example, our model reported that incidence of kidney cancer for males in major cities could be higher than that of regional areas with a probability of 0.605, which still leaves a substantial 30% probability that regional areas do have higher incidence (as was reported by the AIHW [38]).

The remoteness patterns from our proposed meta-analysis model for Australia are also similar to that reported in the Atlas of Cancer in Queensland [57]. The authors considered 13 cancers and four geographical regions: major cities, inner regional, outer regional and remote areas. Cancers having higher incidence in remote areas were lung cancer, oesophageal cancer, cervical cancer; cancers having higher incidence in major cities were breast cancer, kidney cancer; and cancers which have higher incidence in regional (inner or outer) areas were melanoma, prostate and non-Hodgkin lymphoma.

The output of the proposed meta-analysis model was also validated by modelling the observed areal incidence data from 2148 SA2s around Australia. The meta-analysis approach was able to unveil the original regional patterns of cancer incidence found by the unit record model in the majority of cases.

While insights about the different patterns of cancer incidence by remoteness of regions have been widely published [38,57], the rationale of using this methodology to regenerate these patterns was to provide evidence supporting the validation of the proposed modelling approach. The purpose was to confirm whether the proposed model could reveal the same patterns as those observed using the raw data so that this methodology can later be applied to obtain additional insights into other ecological factors which are not already known.

There are limitations to the proposed approach. First, the posterior mean of overall average of each cancer using the proposed meta-analysis model is sometimes different from the Australian average 1 (for lung cancer: 1.05 for males, 1.01 for females, thyroid cancer: males 0.901, females 0.814). However these differences are not substantial as the 95% credible interval around the overall fitted mean contains 1 in all the cases. The reason behind the mean of the parameter not being exactly equal to the Australian average is that, in our proposed meta-analysis model, we are modelling the SIRs, which are ratios themselves and the average of the ratios and the ratio of averages will not be the same [58].

Second, to be able to apply the proposed methodology to online atlases, both point estimates as well as interval estimates or uncertainty estimates need to be available. Moreover, the proposed model might not provide accurate numerical estimates in the presence of outliers; this may explain the more limited validation of the proposed model for remote areas.

We also acknowledge that the proposed methodology could be subject to Simpson's paradox [59–61], which refers to patterns observed for groups of data reversing or disappearing when combined together, since we are analysing summary statistics, as opposed to individual-level data [62,63]. In the spatial context, Simpson's paradox could be considered as a form of modifiable areal unit problem, where results differ depending on the spatial units chosen [64]. Our analysis has the advantage of retaining the original areas within the analysis, and in the case study presented, the areas are the smallest possible available. Results can be considered valid provided they are not extrapolated to an individual level.

The proposed Bayesian hierarchical meta-analysis model is somewhat sensitive to the choice of priors and hyperparameters since they are needed at the lower levels of hierarchy, i.e. for overall mean, overall precision and region-specific precision parameters. Sensible choices of the hyperparameters and prior distributions must be made for the priors of overall mean and precision parameters (or variance parameters) in the model to ensure better model performance. Details of a comprehensive sensitivity analysis to explore this issue are reported in the electronic supplementary material.

The modelling approach proposed in this study can be applied to other published disease atlases that provide point and interval estimates of disease rates. This study explored the behaviour of cancer incidence according to urban/rural status of the small areas. Similar types of analyses can be carried out to identify the influence of different socio-demographic, economic, ecological or environmental factors on specific disease incidence or mortality. This implies that further ecological modelling applying the proposed approach can answer important additional research questions to complement the information already published in the atlas or associated literature. The proposed model can also be extended relatively straightforwardly to a multivariate framework.

Another possible extension of the proposed approach is to combine estimates from other atlases together with ACA and pool the effect of remoteness. There will be some additional challenges in appropriately describing the heterogeneity arising within and between the different studies, and accommodating different definitions of the remoteness categories from the atlases. This is a worthy objective for a future study.

Data accessibility. Two data sources were used. 1. Modelled estimates, freely available and downloaded from: atlas.cancer.org.au/app on 5 October 2018. The URL of the database is: https://atlas.cancer.org.au/app.

After launching the atlas, the top right corner (beside the cancer's name, there is an options button, clicking on that button opens a pop-up for data download. Interested people can type their email in the designated place and the link to download full atlas dataset, in the form of a zip file will be emailed to them within a couple of minutes. This step is necessary for obtaining the data and the Atlas wants to keep record of people who accessed the data.

There are two types of data available in ACA, one on cancer diagnoses and one on cancer survival. The zip files consist of a ReadMe.txt file and four csv files. The ReadMe file explains the data files. In our paper, we used the data file 'SIR Downloadable Data.csv'. This file consists of 2.5%, 10%, 50%, 90% and 97.5% quantile of the SIR for all 20 cancers included in the ACA by sex.

2. Australian Cancer Database, obtained following Ethics and State and Territory Cancer Data Custodian approvals. This dataset is confidential and unavailable, requiring named investigator access through the Secure Unified Research Environment (SURE) facility.

The Australian Cancer Database is managed by the Australian Institute of Health and Welfare (AIHW) and details (including how to request access) are available at: https://www.aihw.gov.au/about-our-data/our-data-collections/australian-cancer-database/about-australian-cancer-database.

Note that this database is a compilation of data from each State and Territory Cancer Registry ($n = 8$), and they remain the data custodians, so obtaining de-identifiable data generally involves the two-step process of ethics approval and public health act approval from each State and Territory Cancer Registry. Contact AIHW first with details of the project to confirm the necessary process. Data is provided as a csv file.

Authors' contributions. F.J. and K.L.M. conceived the modelling approach. F.J. conducted the analyses and led the write-up. E.W.D., S.M.C., P.D.B. and K.L.M. supervised the analyses and interpretations and contributed to the write-up. S.M.C. performed the validation analyses.

Competing interests. The author(s) declared no potential conflicts of interest with respect to the research, authorship and/or publication of this article.

Funding. The author(s) disclosed receipt of the following financial support for the research, authorship and/or publication of this article. This work was supported by Decision Making in the Big Data Era under grant no. FL150100150.

Acknowledgements. The authors acknowledge the work of Australian Cancer Atlas Project Team and funding bodies in developing the atlas. The authors also acknowledge the support of the Australian Research Council (ARC) Centre of Excellence for Mathematical and Statistical Frontiers (ACEMS).

# Appendix A

## A.1. R code for fitting the model using R2jags

```
library(R2jags)
   getSamplesModel <- function(data,inits=NULL, niter = 100000,
   nburnin = 10000, nchains = 3,nthin=10) {

model <- "
model{
for(s in 1:Nsubj){
y[ s] ~dnorm(mu[ s] ,1/sigma_2[ s] )
sigma_2[ s] <-(2*se[ s] _2)/a[ s]
a[ s] ~dchisqr(2)
mu[ s] ~ dnorm(theta[ c[ s]] ,tau[ c[ s]] )
}
for (c in 1:Ncat){
theta[ c] ~dnorm(mu_0,tau_0)
tau[ c] ~dnorm(0,0.1)T(0,)
}
mu_0~dnorm(0,0.01)
tau_0~dnorm(0,0.01)T(0,)
} "
mod <- jags(data,parameters.to.save=c("y","mu","sigma_2","theta",
    "tau", "mu_0","tau_0"),
  model.file = textConnection(model),
  n.chains = nchains, n.iter = niter,
  n.burnin = nburnin, n.thin = nthin)
return(mod)
}
```

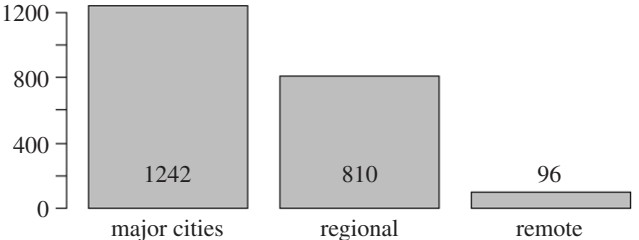

**Figure 11.** Number of areas in different remoteness categories in Australia.

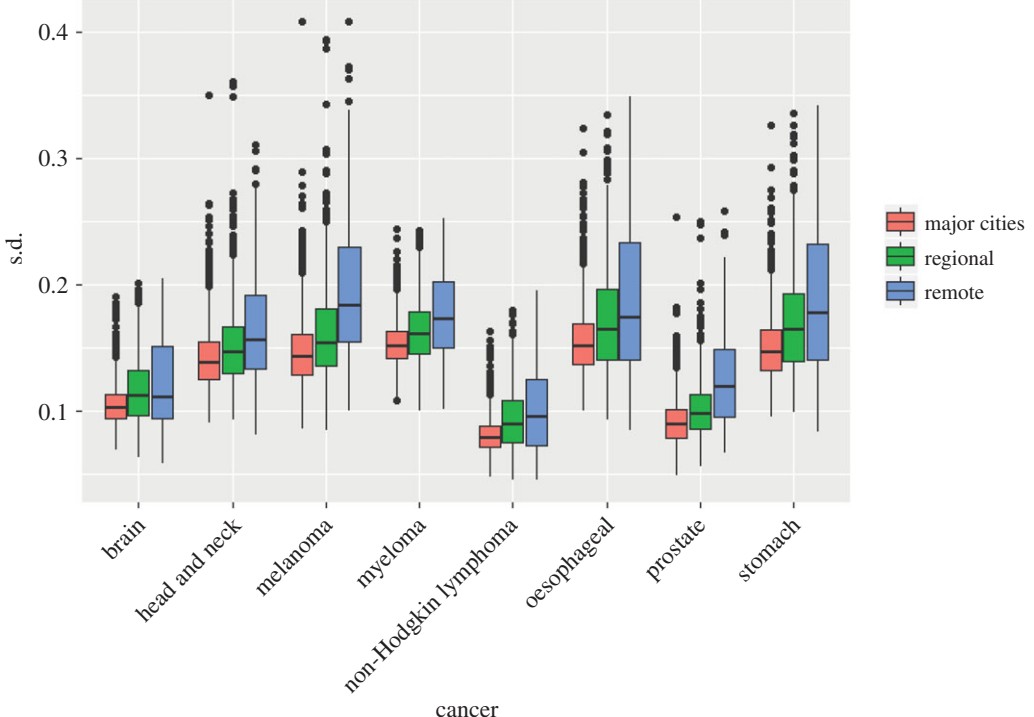

**Figure 12.** Measures of variability among the SIRs of some selected cancers, male.

## A.2. More on larger variation in posterior estimates of remote areas

Apparently, the modelled estimates from the ACA exhibits more variability in remote areas in comparison to major cities and regional areas on the average. We particularly investigated those cancers which showed larger variability in relative differences in estimated SIRs from meta-analysis output and those obtained from analysing the real data. The reasons could be one of the following:

— The smaller number of remote areas than the number of major cities and regional areas among the 2148 small areas (figure 11).
— The larger variability in the estimates of each remote area could be the result of smaller population sizes and fewer cases of some cancers in respective small areas. The inherent variability in estimated SIR in the remote areas for the selected cancers from the ACA makes the credible interval for remote areas wider, which will result in larger relative differences in the posterior means (figures 12 and 13). Perhaps, this can be improved a little by considering the median of the posterior SIRs (from meta-analysis models and model fitted using observed data).

## A.3. Measures of spatial autocorrelation

The spatial autocorrelation among the residuals of the fitted models for all 20 cancers by sex are listed in table 22.

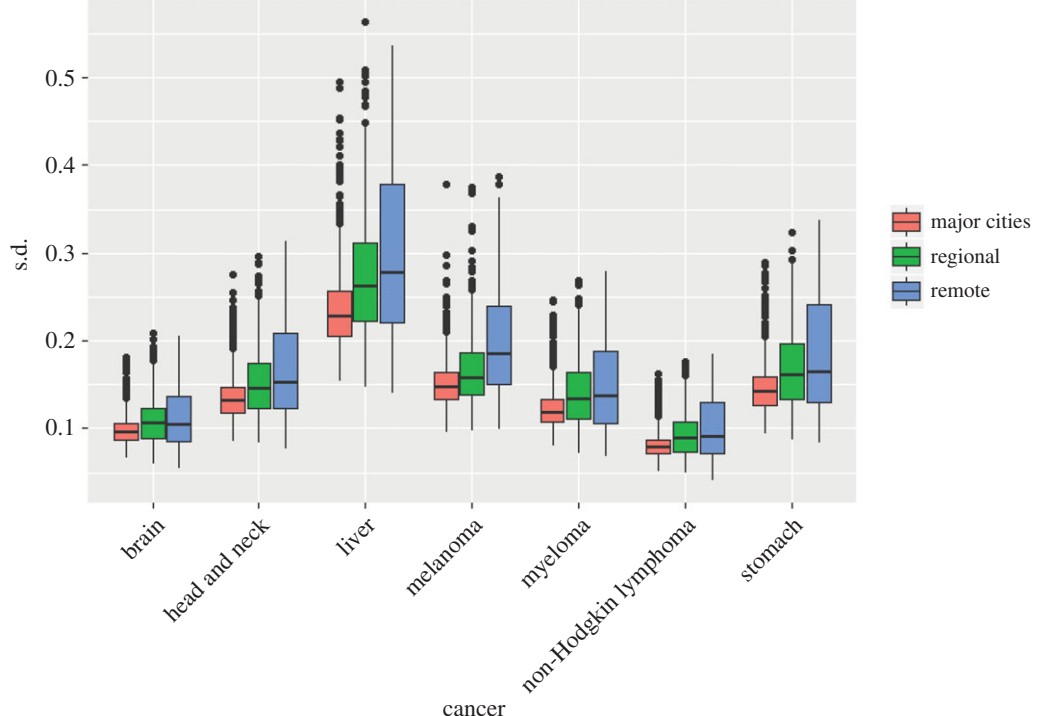

**Figure 13.** Measures of variability among the SIRs of some selected cancers, female.

## A.4. WinBUGS codes for extended model with spatial component

```
model{

for(s in 1:N){
  y[ s] ~dnorm(mu.y[ s], sigma_inv2[ s])
  mu.y[ s] <- mu[ s] + psi[ s]
  sigma_inv2[ s]<-1/sigma_2[ s]
  sigma_2[ s]<-(2 *pow(se[ s],2))/a[ s]
  a[ s] ~dchisqr(2)
  mu[ s] ~ dnorm(theta[ regions[ s]],tau[ regions[ s]])
}
for (j in 1:Ncat){
  theta[ j] ~dnorm(mu_0,tau_0)
  tau[ j] ~dnorm(0,0.1)I(0,)
  }
psi[ 1:N] ~car.normal(adj[], weights[], num[], tau_psi)
mu_0~dnorm(0,0.01)
tau_0~dnorm(0,0.01)I(0,)
tau_psi~dgamma(1.0,0.01)

  }
```

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

**Table 22.** Measures of spatial autocorrelation among the residuals of the fitted models.

| cancer | sex | observed Moran's I | s.d. | p-value[a] |
|---|---|---|---|---|
| all | male | 0.6964 | 0.01449 | <0.0001 |
| | female | 0.7117 | 0.01449 | <0.0001 |
| bowel | male | 0.7113 | 0.01449 | <0.0001 |
| | female | 0.7455 | 0.01449 | <0.0001 |
| brain | male | 0.6783 | 0.01449 | <0.0001 |
| | female | 0.8135 | 0.01448 | <0.0001 |
| breast | female | 0.6789 | 0.01449 | <0.0001 |
| cervical | female | 0.7859 | 0.01450 | <0.0001 |
| head and neck | male | 0.7064 | 0.01449 | <0.0001 |
| | female | 0.7573 | 0.01449 | <0.0001 |
| kidney | male | 0.7784 | 0.01447 | <0.0001 |
| | female | 0.7051 | 0.01449 | <0.0001 |
| leukaemia | male | 0.8812 | 0.01450 | <0.0001 |
| | female | 0.6556 | 0.01449 | <0.0001 |
| liver | male | 0.8948 | 0.01450 | <0.0001 |
| | female | 0.8448 | 0.01450 | <0.0001 |
| lung | male | 0.6644 | 0.01449 | <0.0001 |
| | female | 0.6559 | 0.01449 | <0.0001 |
| melanoma | male | 0.7984 | 0.01449 | <0.0001 |
| | female | 0.8525 | 0.01449 | <0.0001 |
| myeloma | male | 0.4846 | 0.01449 | <0.0001 |
| | female | 0.84847 | 0.01449 | <0.0001 |
| non-Hodgkin lymphoma | male | 0.8166 | 0.01449 | <0.0001 |
| | female | 0.8079 | 0.01449 | <0.0001 |
| oesophageal | male | 0.7592 | 0.01449 | <0.0001 |
| | female | 0.7410 | 0.01450 | <0.0001 |
| ovarian | female | 0.7459 | 0.01449 | <0.0001 |
| pancreatic | male | 0.1538 | 0.01449 | <0.0001 |
| | female | 0.7649 | 0.01449 | <0.0001 |
| prostate | male | 0.7382 | 0.01449 | <0.0001 |
| stomach | male | 0.7688 | 0.01449 | <0.0001 |
| | female | 0.8589 | 0.01450 | <0.0001 |
| thyroid | male | 0.8941 | 0.01450 | <0.0001 |
| | female | 0.8568 | 0.01449 | <0.0001 |
| uterine | female | 0.8144 | 0.01448 | <0.0001 |

[a]Null hypothesis: no spatial autocorrelation is present and the data are randomly distributed.

4. World Health Organization. 2010 *The Atlas of Heart Disease and Stroke*. See https://www.who.int/cardiovascular_diseases/resources/atlas/en/ (accessed 28 April 2019).

5. MRC-PHE Centre for Environment & Health. 2013 *The Environment and Health Atlas for England and Wales*. See http://www.envhealthatlas.co.uk/homepage/ (accessed 28 April 2019).

6. Division of Cancer Epidemiology and Genetics, National Cancer Institute. 2016 *U.S. Atlas of Cancer Mortality*. See https://dceg.cancer.gov/research/how-we-study/descriptive-epidemiology/cancer-mortality-atlas (accessed 8 April 2019).

7. Cancer Council Queensland. 2011 *Queensland Cancer Atlas*. See https://cancerqld.org.au/research/queensland-cancer-statistics/

queensland-cancer-atlas/ (accessed 10 February 2019).

8. Office for National Statistics. 2014 *Cancer Atlas of the United Kingdom and Ireland*. See https://data.gov.uk/dataset/91e37ff6-162e-47ca-8610-50b5c910d94f/cancer-atlas-of-the-united-kingdom-and-ireland (accessed 8 April 2019).

9. Cancer Council Queensland, Queensland University of Technology, Cooperative Research

Centre for Spatial Information. 2018 *Australian Cancer Atlas* (*Version 09-2018*). See http://atlas. cancer.org.au/ (accessed 5 October 2018).

10. Thompson SG, Turner RM, Warn DE. 2001 Multilevel models for meta-analysis, and their application to absolute risk differences. *Stat. Methods Med. Res.* **10**, 375–392. (doi:10.1177/096228020101000602)

11. Kelley GA. 2012 Statistical models for meta-analysis: a brief tutorial. *World J. Methodol.* **2**, 27. (doi:10.5662/wjm.v2.i4.27)

12. Borenstein M, Hedges LV, Higgins JP, Rothstein HR. 2010 A basic introduction to fixed-effect and random-effects models for meta-analysis. *Res. Synth. Methods* **1**, 97–111. (doi:10.1002/jrsm.12)

13. World Health Organization. 2018 *Cancer*. See https://www.who.int/news-room/fact-sheets/detail/cancer.

14. Roquette R, Painho M, Nunes B. 2017 Spatial epidemiology of cancer: a review of data sources, methods and risk factors. *Geospat. Health* **12**, 1–12. (doi:10.4081/gh.2017.504)

15. Bray F, Ferlay J, Soerjomataram I, Siegel RL, Torre LA, Jemal A. 2018 Global cancer statistics 2018: GLOBOCAN estimates of incidence and mortality worldwide for 36 cancers in 185 countries. *CA: A Cancer J. Clinicians* **68**, 394–424. (doi:10.3322/caac.21492)

16. Timander LM, McLafferty S. 1998 Breast cancer in West Islip, NY: a spatial clustering analysis with covariates. *Soc. Sci. Med.* **46**, 1623–1635. (doi:10.1016/S0277-9536(97)10131-9)

17. Xue QY, O'Connell DL, Gibberd RW, Smith DP, Dickman PW, Armstrong BK. 2004 Estimating regional variation in cancer survival: a tool for improving cancer care. *Cancer Causes Control* **15**, 611–618. (doi:10.1023/B:CACO.0000036165.13089.e8)

18. Oliver MN, Smith E, Siadaty M, Hauck FR, Pickle LW. 2006 Spatial analysis of prostate cancer incidence and race in Virginia, 1990–1999. *Am. J. Prev. Med.* **30**, S67–S76. (doi:10.1016/j.amepre.2005.09.008)

19. Cramb SM, Mengersen KL, Baade PD. 2011 Developing the atlas of cancer in Queensland: methodological issues. *Int. J. Health Geogr.* **10**, 9. (doi:10.1186/1476-072X-10-9)

20. Mohebbi M, Wolfe R, Forbes A. 2014 Disease mapping and regression with count data in the presence of overdispersion and spatial autocorrelation: a Bayesian model averaging approach. *Int. J. Environ. Res. Public Health* **11**, 883–902. (doi:10.3390/ijerph110100883)

21. López-Abente G, Aragonés N, Pérez-Gómez B, Pollán M, García-Pérez J, Ramis R, Fernández-Navarro P. 2014 Time trends in municipal distribution patterns of cancer mortality in Spain. *BMC Cancer* **14**, 535. (doi:10.1186/1471-2407-14-535)

22. Dasgupta P, Cramb SM, Aitken JF, Turrell G, Baade PD. 2014 Comparing multilevel and Bayesian spatial random effects survival models to assess geographical inequalities in colorectal cancer survival: a case study. *Int. J. Health Geogr.* **13**, 36. (doi:10.1186/1476-072X-13-36)

23. Cramb SM, Mengersen KL, Lambert PC, Ryan LM, Baade PD. 2016 A flexible parametric approach to examining spatial variation in relative survival. *Stat. Med.* **35**, 5448–5463. (doi:10.1002/sim.7071)

24. Boscoe FP, Ward MH, Reynolds P. 2004 Current practices in spatial analysis of cancer data: data characteristics and data sources for geographic studies of cancer. *Int. J. Health Geogr.* **3**, 28. (doi:10.1186/1476-072X-3-28)

25. Al-Ahmadi K, Al-Zahrani A. 2013 Spatial autocorrelation of cancer incidence in Saudi Arabia. *Int. J. Environ. Res. Public Health* **10**, 7207–7228. (doi:10.3390/ijerph10127207)

26. Bailey T, Hewson P. 2011 Mixtures of factor models for multi-variate disease rates. *Revstat. Stat. J.* **9**, 99–114.

27. Best N, Richardson S, Thomson A. 2005 A comparison of Bayesian spatial models for disease mapping. *Stat. Methods Med. Res.* **14**, 35–59. (doi:10.1191/0962280205sm388oa)

28. Clayton D, Kaldor J. 1987 Empirical Bayes estimates of age-standardized relative risks for use in disease mapping. *Biometrics* **43**, 671–681. (doi:10.2307/2532003)

29. Lee D. 2011 A comparison of conditional autoregressive models used in Bayesian disease mapping. *Spat. Spatiotemporal Epidemiol.* **2**, 79–89. (doi:10.1016/j.sste.2011.03.001)

30. Besag J, York J, Mollie A. 1991 Bayesian image restoration with two applications in spatial statistics. *Ann. Inst. Stat. Math.* **43**, 1–59. (doi:10.1007/BF00116466)

31. Leroux BG, Lei X, Breslow N. 2000 Estimation of disease rates in small areas: a new mixed model for spatial dependence. In *Statistical models in epidemiology, the environment, and clinical trials* (eds ME Halloran, D Berry), pp. 179–191. Berlin, Germany: Springer.

32. Stern HS, Cressie N. 2000 Posterior predictive model checks for disease mapping models. *Stat. Med.* **19**, 2377–2397. (doi:10.1002/1097-0258(20000915/30)19:17/18<2377::AID-SIM576>3.0.CO;2-1)

33. Buja A, Mastrangelo G, Perissinotto E, Grigoletto F, Frigo AC, Rausa G, Marin V, Canova C, Dominici F. 2006 Cancer incidence among female flight attendants: a meta-analysis of published data. *J. Womens Health* **15**, 98–105. (doi:10.1089/jwh.2006.15.98)

34. Baker PJ, Hoel D. 2007 Meta-analysis of standardized incidence and mortality rates of childhood leukaemia in proximity to nuclear facilities. *Eur. J. Cancer Care (Engl).* **16**, 355–363. (doi:10.1111/j.1365-2354.2007.00679.x)

35. Catts V, Catts S, O'Toole B, Frost A. 2008 Cancer incidence in patients with schizophrenia and their first-degree relatives–a meta-analysis. *Acta Psychiatr. Scand.* **117**, 323–336. (doi:10.1111/j.1600-0447.2008.01163.x)

36. Ben Q, Xu M, Ning X, Liu J, Hong S, Huang W, Zhang H, Li Z. 2011 Diabetes mellitus and risk of pancreatic cancer: a meta-analysis of cohort studies. *Eur. J. Cancer* **47**, 1928–1937. (doi:10.1016/j.ejca.2011.03.003)

37. Australian Bureau of Statistics. 2011 'Australian Statistical Geography Standard (ASGS): volume 1-main structure and greater capital city statistical areas'. See https://www.abs.gov.au/websitedbs/D3310114.nsf/home/Australian+Statistical+Geography+Standard+(ASGS).

38. Australian Institute of Health and Welfare. 2016 *Cancer Incidence and Mortality Across Regions (CIMAR) books*. See https://www.aihw.gov.au/reports/cancer/cimar-books/contents/cimar-books (accessed 15 May 2019).

39. Australian Institute of Health and Welfare. 2019 'Cancer in Australia 2019'. See https://www.aihw.gov.au/reports/cancer/cancer-in-australia-2019/contents/summary.

40. Chondur R, Li SQ, Guthridge S, Lawton P. 2014 Does relative remoteness affect chronic disease outcomes? Geographic variation in chronic disease mortality in Australia, 2002–2006. *Aust. N Z J. Public Health* **38**, 117–121. (doi:10.1111/1753-6405.12126)

41. Cramb SM, Mengersen KL, Baade PD. 2011 Identification of area-level influences on regions of high cancer incidence in Queensland, Australia: a classification tree approach. *BMC Cancer* **11**, 311. (doi:10.1186/1471-2407-11-311)

42. Baade P, Dasgupta P, Aitken J, Turrell G. 2011 Geographic remoteness and risk of advanced colorectal cancer at diagnosis in Queensland: a multilevel study. *Br. J. Cancer* **105**, 1039–1041. (doi:10.1038/bjc.2011.356)

43. Heathcote KE, Armstrong BK. 2007 Disparities in cancer outcomes in regional and rural Australia. In *Cancer forum*, vol. 31, p. 70. The Cancer Council Australia.

44. Fox P, Boyce A. 2014 Cancer health inequality persists in regional and remote Australia. *Med. J. Aust.* **201**, 445–446. (doi:10.5694/mja14.01217)

45. Youlden DR, Baade PD, Valery PC, Hassall TE, Ward LJ, Green AC, Aitken JF. 2012 Area-based differentials in childhood cancer incidence in Australia, 1996–2006. *Pediatric Blood & Cancer* **58**, 390–394. (doi:10.1002/pbc.23115)

46. Yu XQ, Luo Q, Smith DP, O'Connell DL, Baade PD. 2014 Geographic variation in prostate cancer survival in New South Wales. *Med. J. Aust.* **200**, 586–590. (doi:10.5694/mja13.11134)

47. DuMouchel W. 1994 Hierarchical Bayes linear models for meta-analysis. Technical report. See https://people.eecs.berkeley.edu/~russell/classes/cs294/f05/papers/dumouchel-1994.pdf.

48. Koricheva J, Gurevitch J, Mengersen K. 2013 *Handbook of meta-analysis in ecology and evolution*. Princeton, NJ: Princeton University Press.

49. Gelman A, Hill J. 2006 *Data analysis using regression and multilevel/hierarchical models*. Cambridge, UK: Cambridge University Press.

50. Gelman A, Rubin DB. 1992 Inference from iterative simulation using multiple sequences. *Stat. Sci.* **7**, 457–472. (doi:10.1214/ss/1177011136)

51. R Core Team. 2018 *R: A Language and Environment for Statistical Computing*. Vienna, Austria: R Foundation for Statistical Computing. See https://www.R-project.org/.

52. Su YS, Yajima M. 2015 R2jags: Using R to run 'JAGS'. R package version 0.5-7. See https://CRAN.R-project.org/package=R2jags.

53. Plummer M, Best N, Cowles K, Vines K. 2006 CODA: convergence diagnosis and output analysis for MCMC. *R News* **6**, 7–11.

54. Hystad P, Carpiano RM, Demers PA, Johnson KC, Brauer M. 2013 Neighbourhood socioeconomic status and individual lung cancer risk: evaluating long-term exposure measures and mediating mechanisms. *Social Sci. Med.* **97**, 95–103. (doi:10.1016/j.socscimed.2013.08.005)

55. Price JH, Everett SA. 1994 Perceptions of lung cancer and smoking in an ecomomically disadvantaged population. *J. Community Health* **19**, 361–375. (doi:10.1007/BF02260405)

56. Cramb SM, Baade PD, White NM, Ryan LM, Mengersen KL. 2015 Inferring lung cancer risk factor patterns through joint Bayesian spatio-temporal analysis. *Cancer Epidemiol.* **39**, 430–439. (doi:10.1016/j.canep.2015.03.001)

57. Cramb SM, Mengersen KL, Baade PD. 2011 Atlas of Cancer in Queensland: geographical variation in incidence and survival, 1998 to 2007. In *Viertel Centre for Res in Cancer Control, Cancer Counc Qld. Brisbane, Qld*. See https://cancerqld.org.au/research/queensland-cancer-statistics/queensland-cancer-atlas/.

58. Larivière V, Gingras Y. 2011 Averages of ratios vs. ratios of averages: an empirical analysis of four levels of aggregation. *J. Informetr.* **5**, 392–399. (doi:10.1016/j.joi.2011.02.001)

59. Blyth CR. 1972 On Simpson's paradox and the sure-thing principle. *J. Am. Stat. Assoc.* **67**, 364–366. (doi:10.1080/01621459.1972.10482387)

60. Scarsini M, Spizzichino F. 1999 Simpson-type paradoxes, dependence, and ageing. *J. Appl. Probab.* **36**, 119–131. (doi:10.1017/S0021900200016892)

61. Hernán MA, Clayton D, Keiding N. 2011 The Simpson's paradox unraveled. *Int. J. Epidemiol.* **40**, 780–785. (doi:10.1093/ije/dyr041)

62. Hanley JA, Thériault G. 2000 Simpson's paradox in meta-analysis. *Epidemiology* **11**, 613. (doi:10.1097/00001648-200009000-00022)

63. Cates CJ. 2002 Simpson's paradox and calculation of number needed to treat from meta-analysis. *BMC Med. Res. Methodol.* **2**, 1. (doi:10.1186/1471-2288-2-1)

64. Cressie NA. 1996 Change of support and the modifiable areal unit problem. *Geograph. Syst.* **3**, 159–180.
