## [Reviewer comments · Royal Society Open Science]

Review History

RSOS-192151.R0 (Original submission)

Review form: Reviewer 1

Is the manuscript scientifically sound in its present form?

Yes

Are the interpretations and conclusions justified by the results?

Yes

Is the language acceptable?

Yes

Do you have any ethical concerns with this paper?

No

Have you any concerns about statistical analyses in this paper?

No

Recommendation?

Major revision is needed (please make suggestions in comments)

Comments to the Author(s)

See attached file (Appendix A).

Review form: Reviewer 2**Is the manuscript scientifically sound in its present form?**

Yes

Are the interpretations and conclusions justified by the results?

Yes

Is the language acceptable?

Yes

Do you have any ethical concerns with this paper?

No

Have you any concerns about statistical analyses in this paper?

Yes

Recommendation?

Major revision is needed (please make suggestions in comments)

Comments to the Author(s)

See attached file (Appendix B).

Decision letter (RSOS-192151.R0)

Dear Mrs Jahan,

The editors assigned to your paper ("Augmenting Disease Maps: a Bayesian meta-analysis approach") have now received comments from reviewers. We would like you to revise your paper in accordance with the referee and Associate Editor suggestions which can be found below (not including confidential reports to the Editor). Please note this decision does not guarantee eventual acceptance.

Please submit a copy of your revised paper before 20-May-2020. Please note that the revision deadline will expire at 00.00am on this date. If we do not hear from you within this time then it will be assumed that the paper has been withdrawn. In exceptional circumstances, extensions may be possible if agreed with the Editorial Office in advance. We do not allow multiple rounds of revision so we urge you to make every effort to fully address all of the comments at this stage. If deemed necessary by the Editors, your manuscript will be sent back to one or more of the original reviewers for assessment. If the original reviewers are not available, we may invite new reviewers.

- Data accessibility

If you wish to submit your supporting data or code to Dryad (<http://datadryad.org/>), or modify your current submission to dryad, please use the following link:
<http://datadryad.org/submit?journalID=RSOS&manu=RSOS-192151>

- Competing interests

- Authors' contributions

AB carried out the molecular lab work, participated in data analysis, carried out sequence alignments, participated in the design of the study and drafted the manuscript; CD carried out the statistical analyses; EF collected field data; GH conceived of the study, designed the study,

coordinated the study and helped draft the manuscript. All authors gave final approval for publication.

- Acknowledgements

- Funding statement

on behalf of Mark Chaplain (Subject Editor)
openscience@royalsociety.org

Associate Editor's comments:
Comments to the Author:

Two reviewers have provided you with substantial commentary. Based on their reports, we would like you to revise the paper to take their concerns into consideration - please ensure you fully respond to their concerns, as it is not generally permissible for subsequent rounds of revision to be offered.

Reviewers' Comments to Author:
Reviewer: 1

Comments to the Author(s)
See attached file (ReviewerComments.pdf)

Reviewer: 2

Comments to the Author(s)
See attached file. (review_RSOS-192151.pdf)

Author's Response to Decision Letter for (RSOS-192151.R0)

See Appendix C.

RSOS-192151.R1 (Revision)

Review form: Reviewer 1

Is the manuscript scientifically sound in its present form?

Yes

Are the interpretations and conclusions justified by the results?

Yes

Is the language acceptable?

Yes

Do you have any ethical concerns with this paper?

No

Have you any concerns about statistical analyses in this paper?

No

Recommendation?

Accept as is

Comments to the Author(s)

The authors addressed all my comments

Review form: Reviewer 2

Is the manuscript scientifically sound in its present form?

Yes

Are the interpretations and conclusions justified by the results?

Yes

Is the language acceptable?

Yes

Do you have any ethical concerns with this paper?

No

Have you any concerns about statistical analyses in this paper?

Yes

Recommendation?

Major revision is needed (please make suggestions in comments)

Comments to the Author(s)

The authors have taken into account most of my comments. However, see comments made to the editor copied below:

As I requested in my first review, the authors have included in section 5.(a) an analysis of the residuals of the fitted models and this shows high spatial autocorrelation. Hence, fitting a liner

regression model without accounting for the spatial variation may lead to the estimates of the fixed effects to be biased.

For this reason, I believe that simply including section 5.(a) is not enough. I understand that including spatial random effects in all the models may take a lot of work but, perhaps, the authors can fit the models with spatial random effects for the models with Moran's I of the residuals equal to 0.15380 and 0.89485, as reported in Section 5.(a). This can be mentioned in Section 5.(a) and included as supplementary materials. This way the authors and readers will have a guidance of the impact of not accounting for the presence of spatial effects in the model and how this affects fixed-effects estimates.

Decision letter (RSOS-192151.R1)

Dear Mrs Jahan:

On behalf of the Editors, I am pleased to inform you that your Manuscript RSOS-192151.R1 entitled "Augmenting Disease Maps: a Bayesian meta-analysis approach" has been accepted for publication in Royal Society Open Science subject to minor revision in accordance with the referee suggestions. Please find the referees' comments at the end of this email.

The reviewers and Subject Editor have recommended publication, but also suggest some minor revisions to your manuscript. Therefore, I invite you to respond to the comments and revise your manuscript.

- Ethics statement

- Data accessibility

<http://datadryad.org/submit?journalID=RSOS&manu=RSOS-192151.R1>

- Competing interests

- Authors' contributions

- Acknowledgements

- Funding statement

Because the schedule for publication is very tight, it is a condition of publication that you submit the revised version of your manuscript before 19-Jun-2020. Please note that the revision deadline will expire at 00.00am on this date. If you do not think you will be able to meet this date please let me know immediately.

- 1) A text file of the manuscript (tex, txt, rtf, docx or doc), references, tables (including captions) and figure captions. Do not upload a PDF as your "Main Document".
- 2) A separate electronic file of each figure (EPS or print-quality PDF preferred (either format should be produced directly from original creation package), or original software format)
- 3) Included a 100 word media summary of your paper when requested at submission. Please ensure you have entered correct contact details (email, institution and telephone) in your user account

4) Included the raw data to support the claims made in your paper. You can either include your data as electronic supplementary material or upload to a repository and include the relevant doi within your manuscript

5) All supplementary materials accompanying an accepted article will be treated as in their final form. Note that the Royal Society will neither edit nor typeset supplementary material and it will be hosted as provided. Please ensure that the supplementary material includes the paper details where possible (authors, article title, journal name).

on behalf of Mark Chaplain (Subject Editor)
openscience@royalsociety.org

Associate Editor Comments to Author:

Comments to the Author:

Thank you for so positively responding to the queries of the reviewers in your initial submission. An outstanding query has been noted that would add clarity to your work. While the reviewer concedes it may require some effort, they argue that the value added would be worthwhile. As the journal recognises that A) we're not living in 'normal' times and B) you may need a slightly longer window to make these changes, if you require an extra week or so to make the change to your manuscript, please let the editorial office know and they will approve this for you, but we would like you to incorporate this change if possible.

Reviewer comments to Author:

Reviewer: 1

Comments to the Author(s)

The authors addressed all my comments

Reviewer: 2

Comments to the Author(s)

The authors have taken into account most of my comments. However, see comments made to the editor copied below:

As I requested in my first review, the authors have included in section 5.(a) an analysis of the residuals of the fitted models and this shows high spatial autocorrelation. Hence, fitting a liner

regression model without accounting for the spatial variation may lead to the estimates of the fixed effects to be biased.

For this reason, I believe that simply including section 5.(a) is not enough. I understand that including spatial random effects in all the models may take a lot of work but, perhaps, the authors can fit the models with spatial random effects for the models with Moran's I of the residuals equal to 0.15380 and 0.89485, as reported in Section 5.(a). This can be mentioned in Section 5.(a) and included as supplementary materials. This way the authors and readers will have a guidance of the impact of not accounting for the presence of spatial effects in the model and how this affects fixed-effects estimates.

Author's Response to Decision Letter for (RSOS-192151.R1)

See Appendix D.

Decision letter (RSOS-192151.R2)

Dear Mrs Jahan,

It is a pleasure to accept your manuscript entitled "Augmenting Disease Maps: a Bayesian meta-analysis approach" in its current form for publication in Royal Society Open Science.

There are two minor items I'd like you to address, please:

1. Your colleague Earl Duncan's email address (earl.duncan@qut.edu.au) appears to be non-functional. Please can you supply us with a correct address?
2. You have provided electronic supplements in both PDF and TeX format. Please can you confirm that both sets of ESM are to be included with the published paper or will one format suffice?

on behalf of Prof Mark Chaplain (Subject Editor)
openscience@royalsociety.org

Appendix A

Review RSOS-192151

General comments

The manuscript by Jahan et al. presents a new methodology for analysing the ecological effect of given risk factors, using published (model-based data) from existing atlases using a Bayesian meta-analysis approach. They compare the proposed methodology with a commonly used ecological regression model adjusting for spatial autocorrelation. They then apply their methodology on 20 types of cancers using smoothed standardized incidence ratios (SIR) as retrieved from the online Australian Cancer Atlas (ACA). The primary goal of the example was to examine the effect of the degree of urbanicity of an area on cancer incidence. They report that their methodology performs fairly similar with the traditional ecological approach, and show the clear distinction of cancer specific SIR with respect to the levels of urbanicity of the areas. Data confidentiality considerations and the increasing amount of online model-based estimates (through atlases or other sources) have raised the need of methods, as the proposed one, that combine and exploit many available data sources. Overall, I find the methodology important as a first step for using model-based estimates and propagating their uncertainty. I have some suggestions that I believe will improve the paper but in general I believe it is well written and well reported.

Major comments

1. You should add a paragraph on the introduction discussing the link of urbanicity and cancer and what the reader should expect from the analysis. Information about the literature, potential hypotheses and reasons for this would make the applications more meaningful. For lung and Thyroid cancer such a discussion is also necessary.
2. Figures 5 and 6 on the appendix are very busy. For a better insight, it would be nice to have these pictures for lung and thyroid cancer and provide a short discussion about their similarities on the corresponding sections.
3. A potential question that can be addressed is how similar the approach you propose is with the simple approach of using the smoothed SIRs on a simple ecological regression model, given that the uncertainty of the outcome is propagated.
4. Page 16, define Simpson's paradox and provide an example related to cancer and areal data.
5. The use of the term meta-analysis prepares the reader for estimates across different studies. It would be interesting thus to show an extension of the proposed approach using estimates from other atlases together with ACA and pool the effect of urbanicity. Heterogeneity is expected with respect to different definitions (which is something natural to expect in a meta-analysis framework). This can also be the objective of a future study, extending to meta-regression to account for the sources of heterogeneity.
6. The validation metrics you used for comparison with the Leroux model should be defined after the definition of the Leroux model and not on the results section. You should also mention that you calculated rank for the Leroux model too.
7. From the epidemiological perspective it would be interesting to transform the urbanicity specific SIRs to relative risks compared to a baseline, say select major cities as the baseline and get the increase/decrease of the SIR in comparison with the other two different levels.

Minor comments

1. The definition of the covariate: major cities, regional and remote areas can be a bit confusing for the reader, since regional could also refer to a regional SIR estimates for instance. I would suggest to refer to it as level/degree of urbanicity instead.
2. The authors refer to areal based disease counts as unit data. This can also confuse and make the reader think that they are referring to individual level data. I would suggest to rephrase using the term areal disease counts or areal level data.
3. The aim and the methodology should come clearer in the introduction. I would suggest stating clear on the 7th paragraph of page 4 that the approach is not a traditional meta analysis approach using estimates from different publications/studies, rather an approach that quantifies the ecological effect of covariates using model-based estimates.
4. On page 5, the index j can be slightly confusing, since each area i can take only one j label. I thus suggest to state the above clear, or redefine it.
5. On page 5, you also need to specify a sex specific index, or to avoid complicating the notation, mention that the analysis was performed with sex stratification.
6. On the lack of cancer specific index, on page 5, you should mention that the analysis was performed for the 20 different cancers independently.
7. Page 7, line 31. W_{ik} should be lowercase.
8. I would suggest to add more information about the definition of urbanicity. Is it just one time point? If yes which one?
9. Page 15, line 41 mentions 4 levels of urbanicity instead of 3.

Appendix B

Review of paper “Augmenting Disease Maps: a Bayesian meta-analysis approach”

The authors present an interesting paper about exploiting data from published atlases of disease. In particular, they propose a meta-analysis to estimate the incidence of a disease according to different types of regions (urban, regional or remote). In general, the paper is well-written and the methods described can be of interest. However, I would like to make some comments about the papers. See below.

General Comments

- Motivation. I think that the use of the meta-analysis on summary statistics from published atlases needs to be better motivated. The obvious reason is that the analysis conducted in the paper using meta-analysis can be done by the original authors of the atlases, as they will always have the original data.
- Type of covariates. The authors describe a meta-analysis to estimate an effect based on categorical covariates. How can this be extended to continuous covariates? Say, for example, that there is information available on risks factors (temperature, humidity, distance to pollution sources, etc.) at the area level. This is neglected in the paper and I think that, at least, it should be paid some attention in the introduction and/or discussion.
- Page 5, lines 40-42. If the original model included a spatial term, then the estimates Y_{ij} are likely to be spatially correlated. Hence, having a spatial term in the meta-analytic model would make sense as to avoid biased estimates of the fixed effects. I would test (using Moran's I) for spatial autocorrelation in the point estimates. If there is (positive) spatial autocorrelation, then I am afraid that a meta-analytic model with some sort of spatial effect will be required. It may be the case that the spatial distribution of ‘remoteness’ explains part of the spatial pattern in the $\log(\text{SIR})$ values though.
In any case, at least I would discuss these issues in the light of the results presented in the paper. It seems that point estimates are close between the meta-analytic model and the one presented on page 7, Section 3.(b). It may happen that because ‘remoteness’ has only three levels and the effect is computed across a large number of areas that the spatial effect is smoothed out. But this should be tackled in the paper.

Specific Comments

- Page 5, line 46. Using notation Y_{ij} is confusing. ‘i’ refers’ to the area and ‘j’ to remoteness, but these are not crossed variables, i.e., for area ‘i’ there is a single value of remoteness. So, I am not sure the sub-index ‘ij’ is adequate. If this is not so, then this should be better explained in the paper.
- Page 5, line 47. I would not call S_{ij} the standard deviation of Y_{ij} if it is computed as in lines 47-48. I fail to see how this can be an estimate of a standard deviation. I would acknowledge that this is a measure of variability though. In addition, perhaps state that there is a relationship between S_{ij} and σ^2_{ij} stated on the next page, where the priors are described.
- Section 3.(a).(i). As mentioned above, how can this model be extended to consider non-categorical covariates?
- Sections 4.(a).i and 4.(b).ii. I believe that the authors should try to come up with some explanation in epidemiological terms for the estimates of ‘remoteness’ obtained, including references to relevant papers (if available). Also, why are some mismatches between the results in the pairwise comparison and the posterior mean differences? Sometimes, for a

very large or small value of the pairwise probability the credible interval of the posterior means difference contains the zero value.

- Page 13, lines 30-33. I wonder whether these larger differences in remote regions are linked to a smaller sample size in these regions. Could the authors present some analysis on the impact of the sample size (i.e., number of cases) on the variability of the estimates? I think that this will explain why estimates for remote regions are widely spread.

Appendix C

We want to thank both reviewers for their time to review this manuscript and for providing comments and suggestions. Our responses to the reviewers' comments are given below:

Reviewer 1:

General Comments

- Motivation. I think that the use of the meta-analysis on summary statistics from published atlases needs to be better motivated. The obvious reason is that the analysis conducted in the paper using meta-analysis can be done by the original authors of the atlases, as they will always have the original data.

Response: Thank you for this comment. We have made this clearer in the manuscript by adding the following sentences in the Introduction.

The meta-analysis approach was proposed because this opens up other avenues for extracting insights from data when only the summary data are published and the original data are unavailable. This is often the case for health data which are subject to privacy and confidentiality. It is true that the original authors could be approached to undertake the follow-on analyses, but this may not always be possible: for example, they could simply refuse, or not have time or funding to implement the request. Moreover, even if the original data were available, the analysis of primary data often requires some domain knowledge. The proposed approach provides a statistically valid methodology to model the published point estimates, taking into account their associated uncertainty, in a straightforward manner. We show that this can facilitate new insights, in our case an enhanced understanding of the spatial distribution of cancer.

- Type of covariates. The authors describe a meta-analysis to estimate an effect based on categorical covariates. How can this be extended to continuous covariates? Say, for example, that there is information available on risks factors (temperature, humidity, distance to pollution sources, etc.) at the area level. This is neglected in the paper and I think that, at least, it should be paid some attention in the introduction and/or discussion.

Response: Thank you for this comment. We have created a new section before the Discussion called Model Extensions, where a model with continuous covariates is formulated as follows:

In the proposed Bayesian hierarchical meta-analysis model, only one covariate is included as a level of hierarchy, which is Remoteness categories. The model can be modified to include continuous covariates if we replace the θ_j by θ in equation 3.2 as:

$$\theta = X \beta$$

where X is a design matrix comprised of continuous covariates and β is the vector of corresponding regression coefficients. The resulting model equation can thus be rewritten as a meta regression model with Normal priors on the regression coefficients.

- Page 5, lines 40-42. If the original model included a spatial term, then the estimates Y_{ij} are likely to be spatially correlated. Hence, having a spatial term in the meta-analytic model would make sense as to avoid biased estimates of the fixed effects. I would test (using Moran's I) for spatial autocorrelation in the point estimates. If there is (positive) spatial autocorrelation, then I am afraid that a meta-analytic model with some sort of spatial effect will be required. It may be the case that the spatial distribution of 'remoteness' explains part of the spatial pattern in the log(SIR) values though.

In any case, at least I would discuss these issues in the light of the results presented in the paper. It seems that point estimates are close between the meta-analytic model and the one presented on page 7, Section 3.(b). It may happen that because 'remoteness' has only three levels and the effect is computed across a large number of areas that the spatial effect is smoothed out. But this should be tackled in the paper.

Response: Thank you for this comment. We have calculated Moran's I for the residuals of our fitted models for all 20 cancers by sex. The results and discussions are included in the model extensions section in the manuscript as follows:

In the present study, we have modelled spatially smoothed estimates of $\log(SIR)$ from ACA. Since the estimated SIRs were already results of a Bayesian spatial model (with a Leroux prior for the spatial term), we did not use any spatial component in our proposed hierarchical Bayesian Meta-analysis model. Further investigation of the residuals of the fitted models for 20 different cancer types by sex resulted in Moran's I values ranging from 0.15 to 0.89 p-values less than 0.0001 (see Appendix (d)). To improve the model performance, a spatial term could be added to the proposed meta-analysis model and a spatial prior could be chosen for modelling. This would be a straightforward extension to the specified model (equation 3.1 -3.7); instead of equation 3.1 in the present model, we can adopt the following:

$$Y_{i[j]} = \mu_{i[j]} + \epsilon_{i[j]} + \psi_{i[j]}$$

where, $\epsilon_{i[j]} \sim N(0, \sigma_{i[j]}^2)$ is an unstructured error component and $\psi_{i[j]}$ is a spatial random effect having a spatial prior. Any suitable spatial prior could be chosen to model the spatial component.

Specific Comments

- Page 5, line 46. Using notation Y_{ij} is confusing. 'i' refers' to the area and 'j' to remoteness, but these are not crossed variables, i.e., for area 'i' there is a single value of remoteness. So, I am not sure the sub-index 'ij' is adequate. If this is not so, then this should be better explained in the paper.

Response: Thank you for this comment. We have updated the description of the notation used in the Model Specification section and modified the notation to $Y_{i[j]}$ as suggested.

- Page 5, line 47. I would not call S_{ij} the standard deviation of Y_{ij} if it is computed as in lines 47-48. I fail to see how this can be an estimate of a standard deviation. I would acknowledge that this is a measure of variability though. In addition, perhaps state that there is a relationship between S_{ij} and $\sigma_{i[j]}^2$ stated on the next page, where the priors are described.

Response: Thank you for noticing this. There was a typo in the formula for S_{ij} which we have corrected. The relationship between S_{ij} and $\sigma_{i[j]}^2$ is more clearly stated in the revised manuscript according to the suggestion as follows:

The standard deviations $S_{i[j]}$ are used to formulate the prior for the associated variance parameter, $\sigma_{i[j]}^2$, which is shown in equation (3.5).

- Section 3.(a).(i). As mentioned above, how can this model be extended to consider non-categorical covariates?

Response: As described above, a separate section has been added in the paper to talk about possible model extensions including the addition of non-categorical variables.

• Sections 4.(a).i and 4.(b).ii. I believe that the authors should try to come up with some explanation in epidemiological terms for the estimates of 'remoteness' obtained, including references to relevant papers (if available). Also, why are some mismatches between the results in the pairwise comparison and the posterior mean differences? Sometimes, for a very large or small value of the pairwise probability the credible interval of the posterior means difference contains the zero value.

Response: Thank you for this comment. Some references to relevant literature about remoteness categories and cancer have been added in the Introduction with discussion about the epidemiological relationship between remoteness and cancer incidence.

Regarding the mismatches in the probabilities and posterior mean differences, the width of credible intervals constructed for the posterior mean differences obviously reflect the amount of variation in these estimates. While calculating the probabilities, we calculated the proportion of times the fitted mean for major cities is larger than the corresponding means for regional and remote areas, and so on. We did not consider the magnitude of the differences or the uncertainty of the obtained probabilities. We inferred differences between regions based on the credible intervals for the posterior mean differences.

The above explanation has been added in the Results section.

• Page 13, lines 30-33. I wonder whether these larger differences in remote regions are linked to a smaller sample size in these regions. Could the authors present some analysis on the impact of the sample size (i.e., number of cases) on the variability of the estimates? I think that this will explain why estimates for remote regions are widely spread.

Response: Thank you for this insightful comment. As is evident in our results, the modelled estimates from the ACA exhibit more variability in remote areas in comparison to major cities and regional areas on average. We have undertaken additional investigation of those cancers which showed larger variability in relative differences in estimated SIRs based on the meta-analysis output and those obtained from analysing the real data. Details of this investigation have been added to the Appendix of the paper (Appendix C) and a brief discussion has been added to the Results section as follows:

However, the relative differences between the posterior mean estimates for remote are much larger. This is because of the smaller number of remote areas and the greater variability between the estimated SIRs in this category. See Appendix (c) for further exposition.

Reviewer 2:

Major Comments:

1. You should add a paragraph on the introduction discussing the link of urbanicity and cancer and what the reader should expect from the analysis. Information about the literature, potential hypotheses and reasons for this would make the applications more meaningful. For lung and Thyroid cancer such a discussion is also necessary.

Response: Thank you for this comment. The link between urbanicity and cancer, along with the relevant literature, has been added to the Introduction as suggested, followed by a clearer motivation to choose the particular application mentioned below. We have also included a discussion of results for lung and thyroid cancers.

In cancer epidemiology, studies to examine the relationship of cancer incidence/survival/mortality and geographic remoteness is well researched [18,2,21,25]. For example, the relationship between risk of advanced colorectal cancer incidence in Queensland and geographic remoteness was found to be

significant for those diagnosed with colon cancer [5]. A classification tree approach approach has also confirmed a significant association between remoteness and the incidence of several cancer [25]. The cancer disparities in different remoteness categories have also been researched by [35,31,63,64] etc. All the mentioned studies focused on the influence of remoteness on cancer outcomes using population based cancer data. Hence, in present study, this well researched and important research question is chosen to provide additional comparisons by which the validity of the proposed approach can be assessed.

2. Figures 5 and 6 on the appendix are very busy. For a better insight, it would be nice to have these pictures for lung and thyroid cancer and provide a short discussion about their similarities on the corresponding sections.

Response: Thanks for the comment. We added a comparison graph for lung and Thyroid cancers with discussion as suggested in the respective subsections as follows:

Lung Cancer: Figures 1 and 2 show the SIRs of lung cancer (for males and females) from the ACA and the fitted Bayesian hierarchical meta-analysis models. There is larger variability in the SIRs from the atlas compared to those of the fitted model, since the effect of remoteness has been removed from the estimates in our fitted model. The atlas estimates are obtained grouping all small area estimates by remoteness categories without considering remoteness in the model, whereas in the proposed model, the posterior SIRs are the result of a model where remoteness categories are included as a covariate to obtain region-specific SIRs for major cities, regional and remote areas.

Thyroid Cancer: Figures 1 and 2 also depicts the SIRs of thyroid cancer (for males and females) from the ACA and the fitted Bayesian hierarchical meta-analysis models.

3. A potential question that can be addressed is how similar the approach you propose is with the simple approach of using the smoothed SIRs on a simple ecological regression model, given that the uncertainty of the outcome is propagated.

Response: Thank you for this comment. The proposed meta-analysis model can be considered as a form of ecological regression set up as a hierarchical model. Here, there is only one categorical explanatory variable in the model, namely remoteness (major cities, regional and remote areas). If we had included other explanatory variables, especially continuous variables, then an ecological linear regression formulation could have been employed.

4. Page 16, define Simpson's paradox and provide an example related to cancer and areal data.

Response: Thank you for the comment. We have added some more references to the literature about Simpson's paradox, with an example related to cancer data. However, examples of Simpson's paradox and areal and spatial data are still under-researched as mentioned in the paper.

We also acknowledge that the proposed methodology could be subject to Simpson's paradox [11,51,36] which refers to patterns observed for groups of data reversing or disappearing when combined together, since we are analysing summary statistics, as opposed to individual level data [34,19]. In the spatial context, Simpson's paradox could be considered as a form of modifiable areal unit problem, where results differ depending on the spatial units chosen [28]. Our analysis has the advantage of retaining the original areas within the analysis, and in the case study presented, the areas are the smallest possible available. Results can be considered valid provided they are not extrapolated to an individual level.

5. The use of the term meta-analysis prepares the reader for estimates across different studies. It would be interesting thus to show an extension of the proposed approach using estimates from other atlases together with ACA and pool the effect of urbanicity. Heterogeneity is expected with respect to different definitions (which is something natural to expect in a meta-analysis framework). This can also be the objective of a future study, extending to meta-regression to account for the sources of heterogeneity.

Response: Thank you for this comment. This is an excellent point and this is indeed an objective for a future study. We have added some sentences about it in the Discussion as follows:

Another possible extension of the proposed approach is to combine estimates from other atlases together with ACA and pool the effect of remoteness. There will be some additional challenges in appropriately describing the heterogeneity arising within and between the different studies, and accommodating different definitions of the remoteness categories from the atlases. This is a worthy objective for a future study.

6. The validation metrics you used for comparison with the Leroux model should be defined after the definition of the Leroux model and not in the results section. You should also mention that you calculated rank for the Leroux model too.

Response: Thank you for the comment. We made the suggested changes in the paper (moved the definition of relative differences in the section after the Leroux model is defined and also added the sentence about calculating rank from Leroux model) as:

The posterior distribution of ranks of the region specific mean SIRs (for major cities, regional and remote areas) are also calculated for both models.

7. From the epidemiological perspective it would be interesting to transform the urbanicity specific SIRs to relative risks compared to a baseline, say select major cities as the baseline and get the increase/decrease of the SIR in comparison with the other two different levels.

Response: Thank you for the comment. We have added the relative risks for lung and thyroid cancer (Table 6,12) and for all different cancers in (Table 14) using major cities as the baseline.

Minor Comments:

1. The definition of the covariate: major cities, regional and remote areas can be a bit confusing for the reader, since regional could also refer to a regional SIR estimates for instance. I would suggest to refer to it as level/degree of urbanicity instead.

Response: With respect, we would prefer not to change the Remoteness label to degree of urbanicity, because the Remoteness index is already well defined by the Australian Bureau of Statistics (ABS). However, we acknowledge the reviewer's comment that the terms can be confusing, so we have changed the description of the remoteness categories in the Data section and clarified that these categories are created by the ABS.

2. The authors refer to areal based disease counts as unit data. This can also confuse and make the reader think that they are referring to individual level data. I would suggest to rephrase using the term areal disease counts or areal level data.

Response: Thanks for the comment. We corrected this throughout the manuscript. Instead of mentioning unit record data, we mentioned observed (which means real data, not estimated or modelled) areal data.

3. The aim and the methodology should come clearer in the introduction. I would suggest stating clearly on the 7th paragraph of page 4 that the approach is not a traditional meta-analysis approach using estimates from different publications/studies, rather an approach that quantifies the ecological effect of covariates using model-based estimates.

Response: Thanks for the comment. We have added this clarification in the manuscript (Introduction) as follows:

The proposed Bayesian hierarchical meta-analysis model approach is not a traditional application of meta-analysis to estimates from different published studies; rather, it is an application of meta-analysis to estimates from a single study, that quantifies the ecological effect of covariates using model-based estimates.

4. On page 5, the index j can be slightly confusing, since each area i can take only one j label. I thus suggest to state the above clear, or redefine it.

Response: Thanks for the comments. We have redefined the notation as suggested and instead of Y_{ij} , we now use the notation $Y_{i[j]}$.

5. On page 5, you also need to specify a sex specific index, or to avoid complicating the notation, mention that the analysis was performed with sex stratification.

Response: Thanks for the comment. We have added this as suggested in the model specification.

6. On the lack of cancer specific index, on page 5, you should mention that the analysis was performed for the 20 different cancers independently.

Response: Thanks for the comment. We have added this as suggested in the model specification.

7. Page 7, line 31. Wik should be lowercase.

Response: Thanks for the comment. We changed W_{ik} to lowercase w_{ik} accordingly.

8. I would suggest to add more information about the definition of urbanicity. Is it just one time point? If yes which one?

Response: Thanks for the comment. We added more information about the Remoteness categories in the Data section of the manuscript.

The Remoteness index is calculated based on 2011 ASGS boundaries. The Australian Bureau of Statistics produce Remoteness Areas, which divide small areas in Australia into five categories of remoteness based on their relative access to services. These remoteness areas are measured using the Accessibility and Remoteness Index of Australia (ARIA+) developed by the University of Adelaide. The ARIA+ is a purely geographic measure of remoteness, and considers services within the specific area, and the distances to localities with more comprehensive services. These measures were originally defined for SA1s, meaning that when aggregated to form SA2s, some SA2s had multiple remoteness categories. For these SA2s, we assigned one remoteness category based on population proportions.

There are five categories of Remoteness – Major city, Inner regional, Outer regional, Remote and Very remote. We have combined Inner and outer regional to Regional, and Remote and Very Remote to Remote areas. We have updated the description about remoteness in the manuscript to avoid confusion as follows:

To address the research question, we focused on the SIR and information on the remoteness status in 2011 of each SA2. This information on remoteness was obtained using the Remoteness Index provided by the Australian Bureau of Statistics (ABS), which classifies small areas in Australia into five categories of remoteness based on their relative access to services. There are five categories of Remoteness – Major city, Inner regional, Outer regional, Remote and Very remote. The original five categories were combined into 3 classes as: 1 = Major Cities (1242 SA2s), 2 = Inner/Outer Regional (810 SA2s) and 3 = Remote/Very Remote (96 SA2s) in the present study.

9. Page 15, line 41 mentions 4 levels of urbanicity instead of 3.

Response: Thanks for the comment. In the mentioned line, we are talking about the Atlas of Cancer, Queensland. The authors of that atlas considered 4 levels of remoteness/urbanicity (Major cities, inner regional, outer regional and remote areas). Hence it is correct in the manuscript in the mentioned line.

Remoteness defined by the ABS actually is a 5-level index, major cities, inner regional, outer regional, remote and very remote areas. The Atlas of Cancer Queensland used 4 categories, combining the remote and very remote as one category. In our paper, we have combined inner and outer regional as regional areas as well as combined remote and very remote to remote areas. Hence, we have 3 levels of remoteness in our study.

Appendix D

29 June, 2020.

Anita Kristiansen
Editorial Coordinator

Dear Concerned,

Thanks for letting me know the decision on the manuscript RSOS-192151.R1 entitled "Augmenting Disease Maps: a Bayesian meta-analysis approach". I have revised the manuscript according to the suggestion from the editor and the reviewer.

The specific revision requested by the reviewer was to fit the proposed model with additional spatial component for two cancers and report the results in Supplementary materials and refer to that in section 5(a). We indeed took this comment positively and we fitted the extended model to liver and pancreatic cancer (males) and reported the results in section 5(a): Model Extensions. We believe, having those output in the main manuscript will increase the credibility of the paper and will give guidance to readers on what has changed by adding the spatial component and when is it necessary.

Other considerations are also made such as: ethics statement, competing interest, data accessibility, authors' contributions, acknowledgements and funding statement in the revised manuscript. I hope the manuscript will be accepted for publication in the Royal Society Open Science.

Please let me know if anything else is needed from my side.

Sincerely yours,

Farzana Jahan
School of Mathematical Sciences,
Science and Engineering Faculty,
Queensland University of Technology,
QLD, Australia.